# SCORE-BASED NEURAL PROCESSES

## ABSTRACT

Neural Processes (NPs) have recently emerged as a powerful meta-learning framework capable of making predictions based on an arbitrary number of context points. However, the learning of NPs and their variants is hindered by the need for explicit reliance on the log-likelihood of predictive distributions, which complicates the training process. To tackle this problem, we introduce Score-based Neural Process (SNP) models, drawing inspiration from recently developed score-based generative models that restore data from noise by reversing a perturbation process. With denoising score matching techniques, the SNPs bypass the intractable log-likelihood calculations, learning parameterized score functions instead. We also demonstrate that score functions possess excellent attributes that enable us to naturally represent a wide family of conditional distributions. Moreover, as data points are inherently unordered, it is crucial to incorporate appropriate inductive biases into SNPs. To this end, we propose building blocks for parameterizing permutation equivariant score functions, which induce the SNPs with the desired properties. Through extensive experimentation on both synthetic and real-world datasets, our SNPs exhibit remarkable performance and outperform existing state-of-the-art NP approaches.

## 1 INTRODUCTION

Meta-learning (Vanschoren, 2018; Thrun & Pratt, 2012) is a promising paradigm that enables networks to acquire suitable priors, thereby improving generalization capability on novel tasks. Neural Processes (NPs) and their variants (Garnelo et al., 2018a;b; Kim et al., 2019) are a set of meta-learning methods that combine the strengths of Gaussian processes (Quinonero-Candela & Rasmussen, 2005) and neural networks to directly model the distribution of functions. As NPs can make predictions based on a limited number of context points while also capturing predictive uncertainty, this allows them to adapt rapidly to novel tasks during testing. As a result, NPs have become a popular choice for a wide range of applications (Kossen et al., 2021; Vaughan et al., 2021; Lin et al., 2021; Garcia-Ortegon et al., 2022; Ada & Ugur, 2023).

Current NP models rely on explicitly maximizing the log-likelihood of target conditional distributions, which requires them to have an analytic form, such as Gaussian densities with a diagonal covariance matrix. However, these approaches do not account for statistical dependencies between the data points, posing challenges in adapting to complex situations and may lead to discontinuous predictions (Dubois et al., 2020; Markou et al., 2022). To address this issue, several NP variants introduce a latent variable into the definition of the predictive distribution, enhancing the model's adaptability to non-Gaussian predictions to some extent (Garnelo et al., 2018b; Kim et al., 2019; Foong et al., 2020). Nevertheless, these variants, similar to other latent variable approaches (Kingma et al., 2019), face the problem of intractable likelihoods with respect to the predictions. In practice, these likelihoods are usually approximated by optimizing a surrogate objective, i.e., evidence lower bound (ELBO) (Kingma et al., 2019). Despite its practical usefulness, it has been observed that optimizing the ELBO of the log-likelihood does not always result in a good latent representation, usually requiring additional modifications to alleviate such problems and improve the quality of latent variables (Alemi et al., 2018; Chen et al., 2016; Wang et al., 2022). Some recent works have introduced an autoregressive structure to deal with hard-to-compute likelihoods (Bruinsma et al., 2023; Nguyen & Grover, 2022) while enhancing the ability to fit the distribution, these models sacrifice the permutation equivariance property of the NP model for target points.

In this paper, we introduce a novel class of neural processes called Score-based Neural Processes (SNPs). Instead of attempting to address the intractable likelihoods with respect to target outputs given

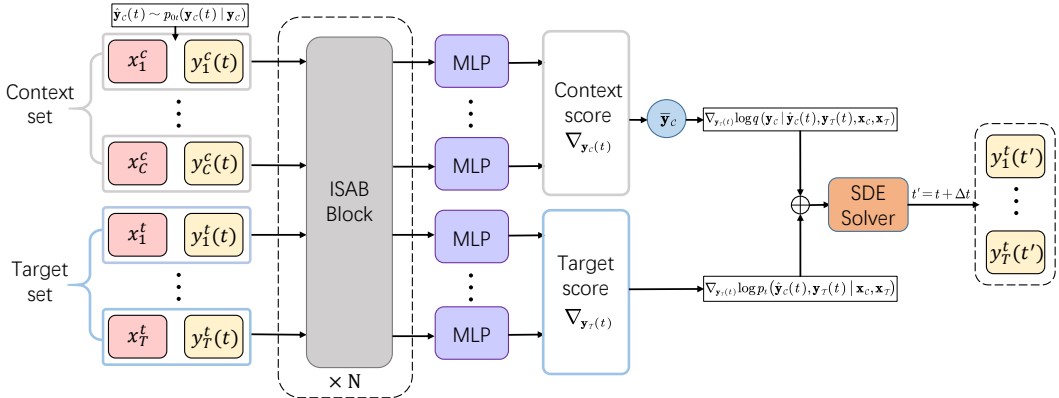

Figure 1: Illustration of score networks for SNPs. At each step, we sample $\hat{\mathbf{y}}_{\mathcal{C}}(t)$ and then use the score network for inference to generate *context score* and *target score*. These scores are applied to estimate $\nabla_{\mathbf{y}_{\mathcal{T}}(t)} \log p_t \left( \mathbf{y}_{\mathcal{T}}(t) \mid \mathbf{x}_{\mathcal{T}}, \mathcal{C} \right)$ by Eq. (7), and the SDE solver is called to predict the target output at the next time step.

context sets, SNPs model score functions (Liu et al., 2016), defined as gradients of log-probability density functions. This allows for more efficient training by score match techniques (Hyvärinen & Dayan, 2005) without relying on the explicit computation of log-likelihoods. Once the score function has been estimated accurately, we can use it to forecast target outputs by solving a reverse-time stochastic differential equation (SDE) (Anderson, 1982). However, learning the score function in a naive manner can be cumbersome since the number of samples in the context set or the target set is variable-sized. This requires the model to include a wide family of conditional distributions. For this purpose, we theoretically show that modeling the score function for the joint distributions of both context and target outputs can effectively be utilized to handle this problem. Additionally, as data points are by nature unordered, it can be crucial to incorporate appropriate inductive biases into the SNPs. We also demonstrate that score functions which are permutation equivariant for data pairs can induce the distribution maintaining these desirable properties, and further offer the practical parameterization to define a score network imposing such constraints.

We conduct extensive experiments on multiple synthetic and real datasets, including meta-regression on synthetic data, multivariate time-series regression on electroencephalogram (EEG) data, and missing data completion on varying domains, such as images on grids, fluid fields on irregular meshes and climate data on manifolds. Our numerical results show that SNPs consistently outperform existing state-of-the-art NP approaches, particularly in more challenging situations, by a large margin. These results demonstrate the ability of SNPs to capture the high-level statistics of related tasks and simulate a wide range of conditional distributions.

## 2 BACKGROUND

### 2.1 PROBLEM SETUP

The meta-learning literature for predictive uncertainty estimation typically involves a meta-dataset $\{\mathcal{D}_k\}_{k=1}^{N_{task}}$ consisting of a series of related tasks. For any task $\mathcal{D}$ sampled from $\{\mathcal{D}_k\}_{k=1}^{N_{task}}$, the dataset is divided into a context set $\mathcal{C} := \{\mathbf{x}_{\mathcal{C}}, \mathbf{y}_{\mathcal{C}}\} = \{(x_i^c, y_i^c)\}_{i=1}^{C}$ and a target set $\mathcal{T} := \{\mathbf{x}_{\mathcal{T}}, \mathbf{y}_{\mathcal{T}}\} = \{(x_i^t, y_i^t)\}_{i=1}^{T}$. Here, $x_i^c \in \mathbb{R}^{d_x}$, $x_i^t \in \mathbb{R}^{d_x}$, $y_i^c \in \mathbb{R}^{d_y}$, $y_i^t \in \mathbb{R}^{d_y}$, $C$ and $T$ are the number of samples for set $\mathcal{C}$ and $\mathcal{T}$ respectively. Our objective is to pursue the conditional distributions $p(\mathbf{y}_{\mathcal{T}} \mid \mathbf{x}_{\mathcal{T}}, \mathcal{C})$, i.e., learning a model which can make reasonable predictions for given target inputs $\mathbf{x}_{\mathcal{T}}$ based on the observed context data $\mathcal{C}$. It's important to note that the number of data points in the context and target sets can vary, necessitating the modeling of a wide family of conditional distributions (Garnelo et al., 2018a).

## 2.2 NEURAL PROCESSES

Neural processes (NPs) (Garnelo et al., 2018b;a; Kim et al., 2019; Vaughan et al., 2021) are meta-learning models, providing an ideal approach for acquiring the above condition distributions. These models are trained by the maximum likelihood procedure that enables fast adaptation to the new task at the testing phase. In general, the recent progress of NPs is mainly driven by two directions: conditional neural processes (CNPs) and latent neural processes (LNPs) (Dubois et al., 2020).

CNPs usually assume that the modeled distribution is factorized conditioned on the context set and can be written by,

$$\log p\left(\mathbf{y}_{\mathcal{T}} \mid \mathbf{x}_{\mathcal{T}}, \mathcal{C}\right) = \sum_{i=1}^{T} \log p\left(y_i^t \mid x_i^t, \mathcal{C}\right) \tag{1}$$

A typical choice is to set each $p\left(y_i^t \mid x_i^t, \mathcal{C}\right)$ as a Gaussian density. CNPs train with ease, however, due to factorization assumption, the major restriction of CNPs is that the model may not be able to produce correlated predictions, resulting in samples that are discontinuous.

LNPs address this problem by introducing a global latent variable. At this point, the likelihood of target prediction is not tractable and LNPs approximately maximize the likelihood by optimizing an evidence lower bound (ELBO):

$$\log p\left(\mathbf{y}_{\mathcal{T}} \mid \mathbf{x}_{\mathcal{T}}, \mathcal{C}\right) \geq \mathbb{E}_{q(z|\mathcal{D})}\left[\sum_{i=1}^{T} \log p\left(y_i^t \mid x_i^t, z\right)\right] - \mathrm{KL}(q(z \mid \mathcal{D}) \| p(z \mid \mathcal{C})) \tag{2}$$

where $z$ is a latent variable incorporated to represent uncertainty. $p(z \mid \mathcal{C})$ is the prior distribution of the latent variable, and $q(z \mid \mathcal{D})$ is the variational posterior distribution based on target sets. Since both the prior and the posterior are unknown, in practice, LNPs usually share the encoder for learning prior and posterior. Yet, this approximation can lead to inference suboptimality (Wang et al., 2022), thereby impairing the performance of models.

## 2.3 SCORE-BASED GENERATIVE MODELS

The recently introduced score-based generative models (SGMs) (Song & Ermon, 2019; 2020; Ho et al., 2020; Song et al., 2020b) offer a flexible sampling approach from high-dimensional, complicated distributions. SGMs first define a diffusion process to perturb data into noise progressively. Considering having a dataset whose samples come from the distribution $p(\mathbf{x})$, SGMs employ the following linear stochastic differential equation (SDE) to perturb samples,

$$d\mathbf{x}(t) = \mu(t)\mathbf{x}(t)dt + \sigma(t)d\mathbf{w}_t, \quad t \in [0, 1], \tag{3}$$

where $\mu(\cdot) : [0, 1] \to \mathbb{R}$, $\sigma(\cdot) : [0, 1] \to \mathbb{R}$, $\{\mathbf{x}(t) \in \mathbb{R}^n\}_{t \in [0,1]}$ denotes the trajectory of the sample being perturbed among the stochastic process, $\mathbf{w}_t$ is a standard Wiener process. Let $p_t(\mathbf{x}(t))$ be marginal probability distribution as time $t$, we apparently have $p_0(\mathbf{x}(0)) \equiv p(\mathbf{x})$. In particular, for some carefully selected $\mu(t)$ and $\sigma(t)$, we can convert an arbitrary initial distribution $p_0(\mathbf{x}(0))$ to a specific noise distribution $p_1(\mathbf{x}(1))$, and the transition kernel $p_{0t}(\mathbf{x}(t) \mid \mathbf{x}(0)) \equiv p_{0t}(\mathbf{x}(t) \mid \mathbf{x})$ at any time during the perturbation process can be obtained in closed-forms. In this paper, we consider the Variance Preserving (VP) SDE proposed in Song et al. (2020b) as the perturbation process, and $p_{0t}(\mathbf{x}(t) \mid \mathbf{x}) = \mathcal{N}(\mathbf{x}(t); \alpha_t\mathbf{x}, \beta_t^2\mathbf{I})$.

SGMs can recover samples $\mathbf{x} \sim p(\mathbf{x})$ from noises by reversing the above perturbation process. A crucial result from Song et al. (2020b) shows that the reverse of a diffusion stochastic process is an SDE that runs backward in time,

$$d\mathbf{x}(t) = \left[\mu(t)\mathbf{x}(t) - \sigma(t)^2 \nabla_{\mathbf{x}(t)} \log p_t\left(\mathbf{x}(t)\right)\right] dt + \sigma(t)d\hat{\mathbf{w}}_t, \quad t \in [0, 1], \tag{4}$$

where $dt$ is the infinitesimal negative timesteps, and $\hat{\mathbf{w}}_t$ is the standard Wiener processes in the reverse-time direction from 1 to 0. The stochastic process in Eq. (4) involves the score function of the marginal distribution, $\nabla_{\mathbf{x}(t)} \log p_t\left(\mathbf{x}(t)\right)$. The score function is typically parameterized as a deep network with the same dimensions of input and output, and can be effectively learned by many score matching techniques (Song et al., 2020a; Vincent, 2011; Pang et al., 2020). Once the score function for all $t$ is given , we can derive the reverse SDE in Eq. (4) and solve it to sample from $p(\mathbf{x})$.

## 3 SCORE-BASED NEURAL PROCESSES

### 3.1 LEARNING SCORE FUNCTIONS FOR NPS

We propose to indirectly achieve meta-learning with neural processes by modeling the score functions instead of the conditional probability densities for target outputs $\mathbf{y}_\mathcal{T}$. For a specific task $\mathcal{D} = \mathcal{C} \cup \mathcal{T}$, where $\mathcal{C} := \{\mathbf{x}_\mathcal{C}, \mathbf{y}_\mathcal{C}\}$ represents the context set and $\mathcal{T} := \{\mathbf{x}_\mathcal{T}, \mathbf{y}_\mathcal{T}\}$ represents the target set, we define a perturbation process $\{\mathbf{y}_\mathcal{T}(t)\}_{t \in [0,1]}$ for the target outputs like in Eq. (3).

$$\mathrm{d}\mathbf{y}_\mathcal{T}(t) = \mu(t)\mathbf{y}_\mathcal{T}(t)\mathrm{d}t + \sigma(t)\mathrm{d}\mathbf{w}_t, \quad t \in [0,1], \tag{5}$$

As time progresses, the target outputs $\mathbf{y}_\mathcal{T} = \mathbf{y}_\mathcal{T}(0)$ will gradually diffuse and finally converge into a simple noise distribution $p_1(\mathbf{y}_\mathcal{T}(1))$. Here, we use the VP SDE as the perturbation process, which results in a standard Gaussian distribution $\mathbf{y}_\mathcal{T}(1) \sim \mathcal{N}(\mathbf{0}, \mathbf{I})$.

Like the stochastic processes described in Eq. (4), we will need to solve the appropriate reverse-time SDE to generate data from the conditional distribution $p(\mathbf{y}_\mathcal{T} \mid \mathbf{x}_\mathcal{T}, \mathcal{C})$,

$$\mathrm{d}\mathbf{y}_\mathcal{T}(t) = \left[\mu(t)\mathbf{y}_\mathcal{T}(t) - \sigma(t)^2 \nabla_{\mathbf{y}_\mathcal{T}(t)} \log p_t\left(\mathbf{y}_\mathcal{T}(t) \mid \mathbf{x}_\mathcal{T}, \mathcal{C}\right)\right]\mathrm{d}t + \sigma(t)\mathrm{d}\hat{\mathbf{w}}_t, \quad t \in [0,1], \tag{6}$$

The above equation involves the conditional score function $\nabla_{\mathbf{y}_\mathcal{T}(t)} \log p_t\left(\mathbf{y}_\mathcal{T}(t) \mid \mathbf{x}_\mathcal{T}, \mathcal{C}\right)$, which is a crucial component but not straightforward to obtain. While one possible approach to model a wide family of conditional distributions is to learn the conditional score function separately for context sets containing various numbers of samples, this can be cumbersome when the range of sample quantity changes is vast. To this end, we propose an alternative way to obtain the score function. We consider the joint distribution with respect to both $\mathbf{y}_\mathcal{T}(t)$ and $\mathbf{y}_\mathcal{C}(t)$, thereby circumventing the need to handle numerous conditional distributions involved by different partitioning of $\mathcal{D}$. This approach offers a more natural and streamlined way to access such conditional score functions. We now state our key theorem.

**Theorem 1.** *Let $\mathcal{D}$ be a task sampled from meta-dataset $\{\mathcal{D}_k\}_{k=1}^{N_{task}}$, which is partitioned into a context set $\mathcal{C} := \{\mathbf{x}_\mathcal{C}, \mathbf{y}_\mathcal{C}\}$ and target set $\mathcal{T} := \{\mathbf{x}_\mathcal{T}, \mathbf{y}_\mathcal{T}\}$. Define the perturbation stochastic processes $\{\mathbf{y}_\mathcal{C}(t)\}_{t \in [0,1]}$ and $\{\mathbf{y}_\mathcal{T}(t)\}_{t \in [0,1]}$ as in Eq. (5). Assume $\hat{\mathbf{y}}_\mathcal{C}(t) \sim p_{0t}(\mathbf{y}_\mathcal{C}(t) \mid \mathbf{y}_\mathcal{C})$. Then, we have the following approximate expression,*

$$\nabla_{\mathbf{y}_\mathcal{T}(t)} \log p_t\left(\mathbf{y}_\mathcal{T}(t) \mid \mathbf{x}_\mathcal{T}, \mathcal{C}\right) \simeq \nabla_{\mathbf{y}_\mathcal{T}(t)} \log p_t\left(\hat{\mathbf{y}}_\mathcal{C}(t), \mathbf{y}_\mathcal{T}(t) \mid \mathbf{x}_\mathcal{C}, \mathbf{x}_\mathcal{T}\right)$$
$$+ \nabla_{\mathbf{y}_\mathcal{T}(t)} \log q(\mathbf{y}_\mathcal{C} \mid \hat{\mathbf{y}}_\mathcal{C}(t), \mathbf{y}_\mathcal{T}(t), \mathbf{x}_\mathcal{C}, \mathbf{x}_\mathcal{T}) \tag{7}$$

The above conclusion is based on Bayes' theorem and approximating expectations by Monte Carlo estimate. We defer the proof in Appendix A. The first term on the right-hand side in Eq. (7) is $\nabla_{\mathbf{y}_\mathcal{T}(t)} \log p_t\left(\hat{\mathbf{y}}_\mathcal{C}(t), \mathbf{y}_\mathcal{T}(t) \mid \mathbf{x}_\mathcal{C}, \mathbf{x}_\mathcal{T}\right)$, which we refer to as the *target score*. Regarding the second term, which necessitates estimating the distribution for $\mathbf{y}_\mathcal{T}$, we define it as a Gaussian density,

$$q(\mathbf{y}_\mathcal{C} \mid \hat{\mathbf{y}}_\mathcal{C}(t), \mathbf{y}_\mathcal{T}(t), \mathbf{x}_\mathcal{C}, \mathbf{x}_\mathcal{T}) = \mathcal{N}(\mathbf{y}_\mathcal{C}; \overline{\mathbf{y}}_\mathcal{C}, r^2\mathbf{I}) \tag{8}$$

where $r$ is the coefficient controlling for variance, and the mean $\overline{\mathbf{y}}_\mathcal{C}$ is the expectation estimated by the result of Tweedie's formula (Robbins, 1992; Chung et al., 2022):

$$\begin{aligned}
\overline{\mathbf{y}}_\mathcal{C} &= \mathbb{E}\left[\mathbf{y}_\mathcal{C} \mid \hat{\mathbf{y}}_\mathcal{C}(t), \mathbf{y}_\mathcal{T}(t), \mathbf{x}_\mathcal{C}, \mathbf{x}_\mathcal{T}\right] \\
&= [\hat{\mathbf{y}}_\mathcal{C}(t) - \beta_t^2 \nabla_{\mathbf{y}_\mathcal{C}(t)} \log p_t\left(\hat{\mathbf{y}}_\mathcal{C}(t) \mid \mathbf{y}_\mathcal{T}(t), \mathbf{x}_\mathcal{C}, \mathbf{x}_\mathcal{T}\right)]/\alpha_t \\
&= [\hat{\mathbf{y}}_\mathcal{C}(t) - \beta_t^2 \nabla_{\mathbf{y}_\mathcal{C}(t)} \log p_t\left(\hat{\mathbf{y}}_\mathcal{C}(t), \mathbf{y}_\mathcal{T}(t) \mid \mathbf{x}_\mathcal{C}, \mathbf{x}_\mathcal{T}\right)]/\alpha_t
\end{aligned} \tag{9}$$

Note that the above equation involves the term $\nabla_{\mathbf{y}_\mathcal{C}(t)} \log p_t\left(\hat{\mathbf{y}}_\mathcal{C}(t), \mathbf{y}_\mathcal{T}(t) \mid \mathbf{x}_\mathcal{C}, \mathbf{x}_\mathcal{T}\right)$, which we refer to as the *context score*. Considering the two terms in Theorem 1 together, our focus is on learning the score function for both $\mathbf{y}_\mathcal{C}(t)$ and $\mathbf{y}_\mathcal{T}(t)$, i.e, $\nabla_{[\mathbf{y}_\mathcal{C}(t), \mathbf{y}_\mathcal{T}(t)]} \log p_t\left(\mathbf{y}_\mathcal{C}(t), \mathbf{y}_\mathcal{T}(t) \mid \mathbf{x}_\mathcal{C}, \mathbf{x}_\mathcal{T}\right)$. To achieve this, we use the denoising score matching (DSM) (Vincent, 2011) objective as our learning criterion,

$$\min_{\boldsymbol{\theta}} \mathbb{E}_{t, \mathcal{D}, \mathbf{y}_\mathcal{C}(t), \mathbf{y}_\mathcal{T}(t)}[\|\mathcal{S}_{\boldsymbol{\theta}}(\mathbf{y}_\mathcal{C}(t), \mathbf{y}_\mathcal{T}(t), \mathbf{x}_\mathcal{C}, \mathbf{x}_\mathcal{T}, t) - \nabla_{[\mathbf{y}_\mathcal{C}(t), \mathbf{y}_\mathcal{T}(t)]} \log p_{0t}\left(\mathbf{y}_\mathcal{C}(t), \mathbf{y}_\mathcal{T}(t) \mid \mathcal{D}\right)\|_2^2]$$
$$\tag{10}$$

where $\mathcal{S}_{\boldsymbol{\theta}}$ is the score network parameterized by $\boldsymbol{\theta}$. The theory of DSM ensures that the optimal solution in Eq. (10) satisfies,

$$\mathcal{S}_{\boldsymbol{\theta}^*}(\mathbf{y}_{\mathcal{C}}(t), \mathbf{y}_{\mathcal{T}}(t), \mathbf{x}_{\mathcal{C}}, \mathbf{x}_{\mathcal{T}}, t) = \nabla_{[\mathbf{y}_{\mathcal{C}}(t), \mathbf{y}_{\mathcal{T}}(t)]} \log p_t(\mathbf{y}_{\mathcal{C}}(t), \mathbf{y}_{\mathcal{T}}(t) \mid \mathbf{x}_{\mathcal{C}}, \mathbf{x}_{\mathcal{T}}) \tag{11}$$

When using the result of Theorem 1, a potential concern is the high variance in estimation, which stems from using single-sample Monte Carlo estimate with $\hat{\mathbf{y}}_{\mathcal{C}}(t) \sim p_{0t}(\mathbf{y}_{\mathcal{C}}(t) \mid \mathbf{y}_{\mathcal{C}})$. Remarkably, this variance is intricately linked to the time progression of the perturbation process within the VP SDE (Song et al., 2020b). The sampling variance reaches its maximum when $t$ is nearly equal to 1 and diminishes as $t$ approaches 0. Previous study (Song & Ermon, 2020; Xu et al., 2022) has affirmed that the generative quality of SGMs primarily hinges on the stage close to $t = 0$, in which the sampling variance is reduced. This characteristic suggests that our approximation in Eq. (7) is reasonable.

In the inference phase, we utilize the trained score network to estimate $\nabla_{\mathbf{y}_{\mathcal{T}}(t)} \log p_t(\mathbf{y}_{\mathcal{T}}(t) \mid \mathbf{x}_{\mathcal{T}}, \mathcal{C})$ by plugging it into Eq. (7) and Eq. (9). These estimated values are then provided to an SDE solver, such as the Euler-Maruyama method (Platen & Bruti-Liberati, 2010), to solve Eq. (6) and generate the target outputs.

## 3.2 Permutation invariance/equivariance

Permutation invariance and equivariance are desirable properties of NP models since the data points in a set are inherently unordered. Integrating these inductive biases into the model parameterization has been verified to be effective for training NP models and is critical for their generalization capacity (Kim et al., 2019). In what follows, we provide the formal definition of these properties.

**Definition 1.** *Context permutation invariance. Let $\Pi_C$ be the set of all permutations of indices $\{1, \ldots, C\}$. If the probability density function of a model satisfy $p(\mathbf{y}_{\mathcal{T}} \mid \mathbf{x}_{\mathcal{T}}, \pi(\mathbf{x}_{\mathcal{C}}), \pi(\mathbf{y}_{\mathcal{C}})) = p(\mathbf{y}_{\mathcal{T}} \mid \mathbf{x}_{\mathcal{T}}, \mathbf{x}_{\mathcal{C}}, \mathbf{y}_{\mathcal{C}})$ for any permutation operator $\pi \in \Pi_C$, then the model is permutation invariance for context sets.*

**Definition 2.** *Target permutation equivariance. Let $\Pi_T$ be the set of all permutations of indices $\{1, \ldots, T\}$. If the probability density function of a model satisfy $p(\pi(\mathbf{y}_{\mathcal{T}}) \mid \pi(\mathbf{x}_{\mathcal{T}}), \mathbf{x}_{\mathcal{C}}, \mathbf{y}_{\mathcal{C}}) = p(\mathbf{y}_{\mathcal{T}} \mid \mathbf{x}_{\mathcal{T}}, \mathbf{x}_{\mathcal{C}}, \mathbf{y}_{\mathcal{C}})$ for any permutation operator $\pi \in \Pi_T$, then the model is permutation equivariance for target sets.*

Definition 1 of context permutation invariance requires that the model's predictions for the target points remain unchanged even when the order of the context points is permuted. Similarly, Definition 2 of target permutation equivariance requires that the predictions $\mathbf{y}_{\mathcal{T}}$ change in accordance with the permutation of the target inputs $\mathbf{x}_{\mathcal{T}}$. In the following, we show that a score network that is permutation equivariant for data pairs can induce the conditional distribution with the aforementioned properties, and this insight will be applied to guide the structural design of our parameterized score networks.

**Theorem 2.** *Assume that $\Pi_{\mathcal{D}}$ is the set of all permutations of indices $\{1, \ldots, C + T\}$. If the score network $\mathcal{S}_{\boldsymbol{\theta}}$ is a permutation equivariant function for data pairs, i.e., satisfies $\mathcal{S}_{\boldsymbol{\theta}}(\pi(\mathbf{y}_{\mathcal{C}}(t), \mathbf{y}_{\mathcal{T}}(t)), \pi(\mathbf{x}_{\mathcal{C}}, \mathbf{x}_{\mathcal{T}}), t) = \pi(\mathcal{S}_{\boldsymbol{\theta}}(\mathbf{y}_{\mathcal{C}}(t), \mathbf{y}_{\mathcal{T}}(t), \mathbf{x}_{\mathcal{C}}, \mathbf{x}_{\mathcal{T}}, t))$, for any choice of $\pi \in \Pi_{\mathcal{D}}$, then the distribution $p_t(\mathbf{y}_{\mathcal{T}}(t) \mid \mathbf{x}_{\mathcal{T}}, \mathcal{C})$ obtained by solving reverse-time SDE in Eq. (6) satisfies the context permutation invariance and target permutation equivariance.*

## 3.3 Architectures of score networks

We aim to design score networks that exhibit permutation equivariance for data pairs. The multihead attention mechanism (Vaswani et al., 2017) is a well-known module that maintains this property while enabling full interaction of information between data points within a set. However, one potential problem is the quadratic time complexity of standard multihead attention, which can become a bottleneck when dealing with a large number of data points. Inspired by sparse Gaussian processes (Snelson & Ghahramani, 2005), we introduce the induced set attention blocks (ISABs) (Lee et al., 2019) as the key components in our score networks. Specifically, given the set of data point $\{\mathbf{y}_{\mathcal{C}}(t), \mathbf{y}_{\mathcal{T}}(t), \mathbf{x}_{\mathcal{C}}, \mathbf{x}_{\mathcal{T}}\}$ at specific time $t$, we first concatenate the corresponding inputs and perturbed outputs together and project them to hidden space $Z \in \mathbb{R}^{(C+T) \times d_h}$. ISABs use $m$ trainable inducing points $I \in \mathbb{R}^{m \times d_h}$ (where $m \ll C + T$) as the query matrices to extract meaningful

low-dimensional feature $H \in \mathbb{R}^{m \times d_h}$ from the input array $Z$ by multihead attention. The output is then generated by reconstructing back to the original space based on $H$. The full process of ISABs can be defined as follows:

$$\text{ISAB}(Z) = Z + \text{FFN}(\text{MultiHead}(Z, H, H)) \in \mathbb{R}^{(C+T) \times d_h}$$

$$\text{where } H = I + \text{FFN}(\text{MultiHead}(I, Z, Z)) + \text{Embed}(t) \in \mathbb{R}^{m \times d_h} \tag{12}$$

where $\text{MultiHead}(\cdot, \cdot, \cdot)$ is a multihead attention module whose inputs are, in order, query-key-value matrices, $\text{FFN}(\cdot)$ is the feed-forward network for mixing on channel dimension, $\text{Embed}(\cdot)$ is the layer embedding the time of SDEs. As the number of inducing points $m$ is far less than $C + T$, this reduces the computational complexity from $\mathcal{O}((C + T)^2)$ to $\mathcal{O}(m(C + T))$, allowing our model to easily process thousands of data points. More importantly, ISABs still maintain the equivariant property for score networks in Theorem 2. We stack multiple ISABs and add a fully-connected output layer at the tail to construct the score networks $\mathcal{S}_\theta$. The overall architecture and the workflow of score calculation are illustrated in Figure 1.

## 4 EXPERIMENTAL RESULTS

We conduct a comprehensive evaluation of Score-based Neural Processes (SNPs) across several tasks, including meta-regression on synthetic data, multivariate time-series regression on electroencephalogram (EEG) data, and missing data completion in various domains, such as images on grids, fluid fields on irregular meshes and climate data on manifolds. We compare SNPs with other NP models, including Neural Processes (NPs) (Garnelo et al., 2018b), Attentive Neural Processes (ANPs) (Kim et al., 2019), and Convolutional Neural Processes (ConvNPs) (Foong et al., 2020), while keeping the number of parameters approximately the same for each model to ensure a fair comparison. We provide implementation details in Appendix C for reference.

### 4.1 SYNTHETIC EXPERIMENTS

For each experiment, we conduct the meta-regression tasks by collecting functions from Gaussian processes (GPs) and sub-sampling these functions into context $\{(x_i^c, y_i^c)\}_{i=1}^C$ and target sets $\{(x_i^t, y_i^t)\}_{i=1}^T$, on which we train the NP models. Specifically, we apply the NP models to synthetic datasets generated from GPs with three following settings. *1) GPs with the single kernel:* we generate functions from GPs with RBF, Periodic and Matern kernels, respectively, to build three meta-datasets. *2) GPs with varying kernels:* we generate functions from GPs with three different kernels and combine them into one meta-dataset. *3) GPs with varying kernel hyperparameters:* we generate functions from GPs with Matern kernels, with a length scale of $l \sim U[0.01, 0.3]$.

Table 1: Predictive NLL ($\downarrow$) on synthetic data (5 runs).

| Method | Single kernel | | | Varying kernels | Varying h.p. |
|--------|------|----------|--------|-----------------|--------------|
| | RBF | Periodic | Matérn | | |
| NP | $1.13 \pm 0.02$ | $0.66 \pm 0.01$ | $1.26 \pm 0.01$ | $1.61 \pm 0.01$ | $1.66 \pm 0.02$ |
| ANP | $0.01 \pm 0.02$ | $0.64 \pm 0.01$ | $0.53 \pm 0.03$ | $0.52 \pm 0.01$ | $0.96 \pm 0.01$ |
| ConvNP | $-0.85 \pm 0.01$ | $-1.88 \pm 0.01$ | $0.32 \pm 0.03$ | $0.39 \pm 0.00$ | $0.95 \pm 0.01$ |
| SNP | $\mathbf{-3.64} \pm 0.00$ | $\mathbf{-4.14} \pm 0.02$ | $\mathbf{-0.98} \pm 0.00$ | $\mathbf{-2.43} \pm 0.01$ | $\mathbf{0.45} \pm 0.00$ |

We test the trained models respectively on unseen functions that are drawn from the above three settings and calculate the negative log-likelihood (NLL) of the target sets. The summarized results are shown in Table 1. We can see that SNPs consistently outperform the other methods in all settings. Particularly in the case of mixing from different kernels, the SNP also has obviously lower NLL, indicating its ability to identify and quickly adapt to the task from different kernels. We refer the readers to Appendix E for more qualitative results.

### 4.2 EEG REGRESSION

Next, we train various NPs on real series data consisting of EEG measurements (Zhang et al., 1995), following the methodology described in Bruinsma et al. (2023). Each time series contains 256

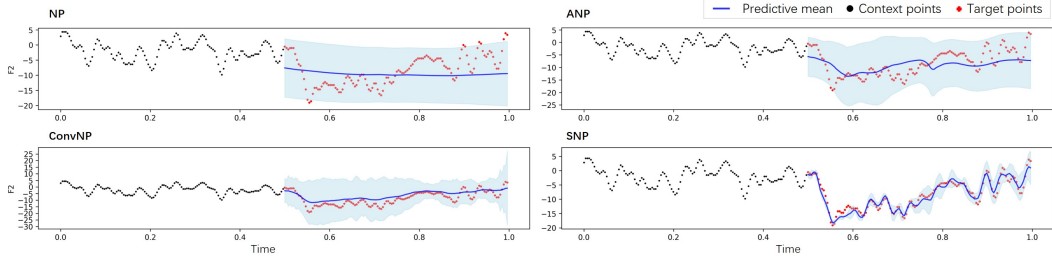

Figure 2: Qualitative evaluation of *Forecasting* on EEG. The blue lines and the shaded blue area denote the predictive mean and standard deviation.

regularly spaced measurements across 7 channels (i.e., FZ, F1, F2, F3, F4, F5, F6 electrodes), and exhibits correlations across channels, making it an ideal multivariate regression task for meta-learning models. The regression's inputs are the time and channel index $x_e := (i_t, i_c) \in \mathbb{R}^2$ and the output is the corresponding voltage $y_e$. We maintain 4 channels as the context set and conducted three experiments on other channels, which are *Interpolation*, *Reconstruction* and *Forecasting*.

Table 2: Predictive NLL ($\downarrow$) and MSE ($\downarrow$) on EEG (5 runs).

| Method | Inter. | | Recon. | | Forec. | |
|--------|--------|--------|--------|--------|--------|--------|
| | NLL | MSE($\times 10^{-2}$) | NLL | MSE($\times 10^{-2}$) | NLL | MSE($\times 10^{-2}$) |
| NP | $1.24 \pm 0.00$ | $0.35 \pm 0.01$ | $1.24 \pm 0.00$ | $0.32 \pm 0.00$ | $1.24 \pm 0.00$ | $0.30 \pm 0.04$ |
| ANP | $0.36 \pm 0.02$ | $0.16 \pm 0.01$ | $0.49 \pm 0.02$ | $0.34 \pm 0.02$ | $0.70 \pm 0.00$ | $0.43 \pm 0.05$ |
| ConvNP | $0.31 \pm 0.02$ | $0.29 \pm 0.01$ | $-1.79 \pm 0.01$ | $0.30 \pm 0.00$ | $-1.78 \pm 0.00$ | $0.41 \pm 0.03$ |
| SNP | $\mathbf{-2.37} \pm 0.00$ | $\mathbf{0.11} \pm 0.01$ | $\mathbf{-2.40} \pm 0.01$ | $\mathbf{0.12} \pm 0.01$ | $\mathbf{-2.35} \pm 0.00$ | $\mathbf{0.22} \pm 0.00$ |

After training, we evaluate the models by calculating the NLL and MSE of the target sets. As shown in Table 2, the SNP outperforms the other methods on all tasks by a large margin, especially in more difficult *Forecasting* tasks. Figure 2 shows some qualitative results of forecasting on the F2 channel. We observe that the trained SNP, which had temporal and cross-channel correlations captured, produces both regression uncertainty and plausible function samples given context channels.

### 4.3 IMAGES ON GRIDS

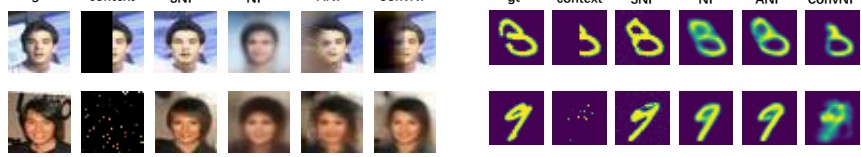

Figure 3: Qualitative evaluation on CelebA32 (left) and MNIST (right).

Image data can be interpreted as being generated from a stochastic process, where each image can be regarded as a unique function. Predicting the pixel values can be cast as a 2D regression task mapping a 2D pixel location $x_i \in \mathbb{R}^2$ to its pixel intensity $y_i \in \mathbb{R}^c$, where $c$ is the number of channels of images. We used two benchmark datasets for this experiment: MNIST (LeCun et al., 1998) and CelebA (Liu et al., 2015). As in Lee et al. (2020), we downsample the CelebA images to $32 \times 32$. The coordinates and pixel intensities are both rescaled to the range $[-1, 1]$. We use the standard train/test split for two image datasets.

During the test phase, we evaluate each model based on the log-likelihood of the target points on the test data. Our results are similar to our previous findings, as shown in Table 3, which demonstrates that SNPs outperform the baselines and achieve a nearly three-fold likelihood improvement. The qualitative results are shown in Figure 3, where we find that SNPs produce noticeably better-completed images than the best baseline model for different context sampling. Even in more challenging tasks, such as contexts containing only half of the image, which require the NP models to simulate dependencies between distant pixels, our model produces more reliable predictions than the baseline,

Table 3: Predictive NLL ($\downarrow$) on MNIST and CelebA32 (5 runs).

| Method | MNIST | | CelabA32 | |
|---|---|---|---|---|
| | Half | Random | Half | Random |
| NP | $-3.63 \pm 0.00$ | $-3.54 \pm 0.01$ | $-0.84 \pm 0.00$ | $-1.07 \pm 0.00$ |
| ANP | $-3.91 \pm 0.00$ | $-3.96 \pm 0.01$ | $-1.39 \pm 0.00$ | $-1.43 \pm 0.01$ |
| ConvNP | $-2.57 \pm 0.84$ | $-2.96 \pm 0.01$ | $0.11 \pm 0.28$ | $-1.67 \pm 0.01$ |
| **SNP** | $\mathbf{-9.12} \pm 0.08$ | $\mathbf{-8.87} \pm 0.01$ | $\mathbf{-4.76} \pm 0.10$ | $\mathbf{-4.77} \pm 0.10$ |

which generates predictions inconsistent with the context. This discrepancy explains why the likelihoods of the baseline models were much lower than those of SNPs.

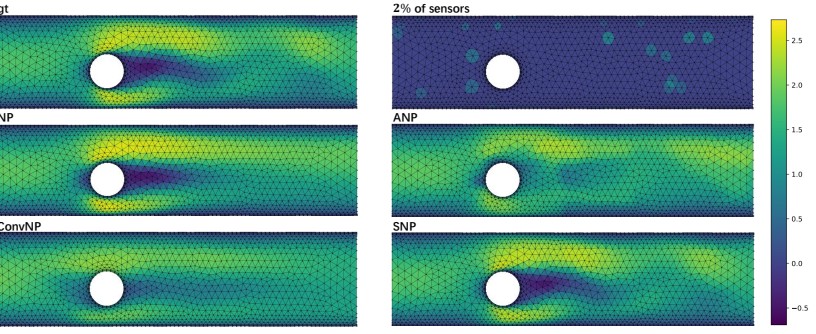

Figure 4: Qualitative evaluation under 2% of sensors on CYLINDERFLOW. The color map shows the x-component of the velocity field.

### 4.4 FLOW FIELDS ON MESHES

Like other NP models, SNPs are not limited to grid data and can theoretically access function values at arbitrary coordinate points. In this subsection, we collect flow field data lying on varying irregular meshes to verify the model's ability to recover the complete system from very sparse observations. This approach can be used to address the variational data assimilation problem (Frerix et al., 2021) in numerical systems that are affected by non-invertible relations between physical states and their corresponding observations. Here, we adopt the CYLINDERFLOW (Pfaff et al., 2020), a fluid simulation dataset consisting of 600 temporal observations per trajectory, with 1735 to 2036 evaluation nodes. Our target is to learn the data distribution of the velocity fields $y_i := (v_i^x, v_i^y) \in \mathbb{R}^2$ for each node, given its position, time (The meaning of time here is different from that in Eq. (6)) and node type, i.e., $x_i := (p_i^x, p_i^y, s_i, n_i) \in \mathbb{R}^4$. Due to the delicate design of the 2D grid version of ConvNPs Gordon et al. (2019), we have to interpolate the irregular meshes to a $32 \times 32$ grid and discard the time information to fit it. For fairness, we only evaluate all NP methods on completing the missing nodes spatially with $2\%$ of sensors.

The results of the mean squared error (MSE) for the target point are summarized in Table 4. Our SNP consistently outperforms the baseline, with an accuracy one order of magnitude higher than the second-best method (i.e., NP). In Figure 4, we compare the completion results of varying NP models. The SNP yields a result that is closer to the ground truth than other models.

Table 4: Test MSEs ($\downarrow$) on CYLINDERFLOW (5 runs).

| | NP | ANP | ConvNP | SNP |
|---|---|---|---|---|
| MSE($\times 10^{-2}$) | $1.08 \pm 0.03$ | $1.57 \pm 0.00$ | $3.41 \pm 0.01$ | $\mathbf{0.11} \pm 0.01$ |

### 4.5 CLIMATE DATA ON MANIFOLDS

Data in earth and climate science often exists on a manifold rather than in Euclidean space, bringing a great modeling challenge. The proposed SNP can be naturally scaled to this data without modifications as in De Bortoli et al. (2022). To verify this, we adopt temperature measurements over the last 40 years from the ERA5 dataset (Hersbach et al., 2019). For NP, ANP

and SNP, we consider learning to predict the temperature value $y_{\theta,\phi}$ given spherical coordinates $x_{\theta,\phi} := (\cos(\theta)\cos(\phi), \cos(\theta)\sin(\phi), \sin(\theta)) \in \mathbb{R}^3$, which can ensure that the data lie on manifolds (Dupont et al., 2021). Since the ConvNPs are not directly applicable to this case, we feed the input in the form of a latitude/longitude grid (similar to image data) into it for training. We conduct two kinds of experiments, including recovery from Large-region missing and random missing.

Table 5: Predictive NLL ($\downarrow$) and MSEs ($\downarrow$) on ERA5 (5 runs).

| Method | Large-region missing | | Random missing | |
|---|---|---|---|---|
| | NLL | MSE($\times 10^{-2}$) | NLL | MSE($\times 10^{-2}$) |
| NP | $-3.35 \pm 0.01$ | $0.14 \pm 0.05$ | $-3.36 \pm 0.00$ | $0.16 \pm 0.00$ |
| ANP | $-4.58 \pm 0.09$ | $0.16 \pm 0.01$ | $-3.85 \pm 0.02$ | $0.08 \pm 0.00$ |
| ConvNP | $4.89 \pm 2.22$ | $10.94 \pm 0.53$ | $0.50 \pm 0.61$ | $0.92 \pm 0.11$ |
| **Ours** | $\mathbf{-5.05 \pm 0.13}$ | $\mathbf{0.05 \pm 0.01}$ | $\mathbf{-5.03 \pm 0.00}$ | $\mathbf{0.06 \pm 0.00}$ |

In Table 5, we observe that the SNP outperforms other competing methods in terms of NLL and MSE for both random missing cases and large-region missing cases. The performance of ConvNP is significantly lower than that of the other models, and we speculate that this may be due to the use of latitude and longitude to represent the location, ignoring the fact that the data comes from manifolds. Moreover, the visualizations in Figure 5 demonstrate the ability of the SNP to extrapolate non-stationary, complex patterns, especially in regions with rapid and discontinuous temperature changes. In comparison, the baseline NP models produce either blurry or unreliable predictions. These results suggest that the SNP can provide more accurate and reliable predictions for applications of earth and climate science. For further details and qualitative results, please refer to Appendix E.

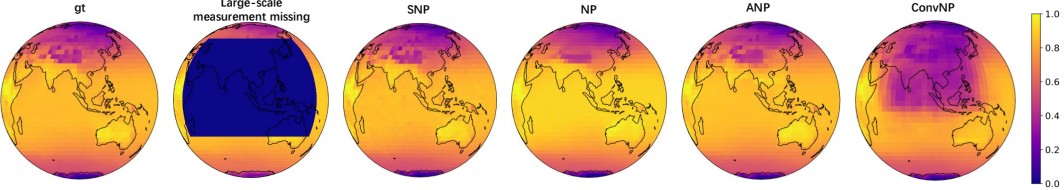

Figure 5: Qualitative evaluation under large-scale measurements missing on ERA5.

## 4.6 ABLATION STUDY

**Assessing the efficacy of ISAB:** To validate the effectiveness of the proposed ISAB, we conducted a comprehensive comparative analysis with vanilla attention (Vaswani et al., 2017), as elaborated in the Appendix D.1. To quantify the computational demands of models, we employed the Multiply ACcumulate operations (MACs) as a metric. Our findings demonstrate that the ISAB block strikes a commendable balance between computational efficiency and precision. In most instances, it outperforms the standard attention mechanism while demanding fewer computational resources, particularly when handling extensive datasets such as ERA5.

**Runtime tradeoff:** The SNP model necessitates additional computational resources due to the inclusion of the SDE solver, setting it apart from other NP models. In the Appendix D.2, we delve into the impact of varying the number of function estimations (NFE), on both runtime and model performance on EEG for SNP. Our findings illustrate a robust improvement in SNP's performance as NFE increases. Impressively, even with a relatively low NFE (e.g., NFE = 50), SNP exhibits a slight advantage over the baseline. The exploration of employing more advanced solvers to enhance SNP's performance while reducing the demand for NFEs presents a promising avenue for future research.

## 5 CONCLUSION

In this paper, we propose SNPs as novel members of the NP family for meta-learning. SNPs model score functions and learn them by using score matching techniques, thereby avoiding dealing with intractable likelihood functions. We demonstrate that SNPs can represent wide, complex conditional distributions. Moving forward, we plan to extend the application of SNPs to higher dimensional meta-learning scenarios. We are also interested in exploring the connections between NPs and in-context learning paradigms (Liu et al., 2021; Brown et al., 2020), and incorporating recent advances in these areas to enhance the performance of SNPs.

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

# A  PROOFS

In this section, we provide proof for the conclusions in the main text.

## A.1  PROOF OF THEOREM 1

$$
\begin{aligned}
p_t\left(\mathbf{y}_\mathcal{T}(t) \mid \mathbf{x}_\mathcal{T}, \mathcal{C}\right) &= \int p_t\left(\mathbf{y}_\mathcal{C}(t), \mathbf{y}_\mathcal{T}(t) \mid \mathbf{x}_\mathcal{T}, \mathcal{C}\right) \mathrm{d}\mathbf{y}_\mathcal{C}(t) \\
&= \int p_t\left(\mathbf{y}_\mathcal{T}(t) \mid \mathbf{y}_\mathcal{C}(t), \mathbf{x}_\mathcal{T}, \mathcal{C}\right) p_{0t}\left(\mathbf{y}_\mathcal{C}(t) \mid \mathbf{y}_\mathcal{C}\right) \mathrm{d}\mathbf{y}_\mathcal{C}(t) \quad (13) \\
&= \mathbb{E}_{p_{0t}(\mathbf{y}_\mathcal{C}(t) \mid \mathbf{y}_\mathcal{C})}\left[p_t\left(\mathbf{y}_\mathcal{T}(t) \mid \mathbf{y}_\mathcal{C}(t), \mathbf{x}_\mathcal{T}, \mathcal{C}\right)\right] \\
&\simeq p_t\left(\mathbf{y}_\mathcal{T}(t) \mid \hat{\mathbf{y}}_\mathcal{C}(t), \mathbf{x}_\mathcal{T}, \mathcal{C}\right) \quad \text{where } \hat{\mathbf{y}}_\mathcal{C}(t) \sim p_{0t}\left(\mathbf{y}_\mathcal{C}(t) \mid \mathbf{y}_\mathcal{C}\right)
\end{aligned}
$$

Then,

$$
\begin{aligned}
\nabla_{\mathbf{y}_\mathcal{T}(t)} \log p_t\left(\mathbf{y}_\mathcal{T}(t) \mid \mathbf{x}_\mathcal{T}, \mathcal{C}\right) &\simeq \nabla_{\mathbf{y}_\mathcal{T}(t)} \log p_t\left(\mathbf{y}_\mathcal{T}(t) \mid \hat{\mathbf{y}}_\mathcal{C}(t), \mathbf{x}_\mathcal{T}, \mathcal{C}\right) \\
&= \nabla_{\mathbf{y}_\mathcal{T}(t)}[\log p_t\left(\hat{\mathbf{y}}_\mathcal{C}(t), \mathbf{y}_\mathcal{T}(t) \mid \mathbf{x}_\mathcal{T}, \mathcal{C}\right) - \log p_t\left(\hat{\mathbf{y}}_\mathcal{C}(t) \mid \mathbf{x}_\mathcal{T}, \mathcal{C}\right)] \\
&= \nabla_{\mathbf{y}_\mathcal{T}(t)} \log p_t\left(\hat{\mathbf{y}}_\mathcal{C}(t), \mathbf{y}_\mathcal{T}(t) \mid \mathbf{x}_\mathcal{T}, \mathcal{C}\right) \\
&= \nabla_{\mathbf{y}_\mathcal{T}(t)} \log p_t\left(\hat{\mathbf{y}}_\mathcal{C}(t), \mathbf{y}_\mathcal{T}(t) \mid \mathbf{x}_\mathcal{C}, \mathbf{x}_\mathcal{T}, \mathbf{y}_\mathcal{C}\right) \quad (14) \\
&\quad (\text{ By Bayes' theorem}) \\
&= \nabla_{\mathbf{y}_\mathcal{T}(t)}[\log p_t\left(\hat{\mathbf{y}}_\mathcal{C}(t), \mathbf{y}_\mathcal{T}(t) \mid \mathbf{x}_\mathcal{C}, \mathbf{x}_\mathcal{T}\right) \\
&\quad + \nabla_{\mathbf{y}_\mathcal{T}(t)} \log q(\mathbf{y}_\mathcal{C} \mid \hat{\mathbf{y}}_\mathcal{C}(t), \mathbf{y}_\mathcal{T}(t), \mathbf{x}_\mathcal{C}, \mathbf{x}_\mathcal{T})]
\end{aligned}
$$

## A.2  PROOF OF THEOREM 2

As the initial noise distribution at time 1 is standard normal distribution, $p_1\left(\mathbf{y}_\mathcal{T}(1) \mid \mathbf{x}_\mathcal{T}, \mathbf{y}_\mathcal{C}, \mathbf{x}_\mathcal{C},\right) = \mathcal{N}(\mathbf{0}, \mathbf{I})$, we have,

$$
p_1\left(\mathbf{y}_\mathcal{T}(1) \mid \mathbf{x}_\mathcal{T}, \pi(\mathbf{x}_\mathcal{C}), \pi(\mathbf{y}_\mathcal{C})\right) = p_1\left(\mathbf{y}_\mathcal{T}(1) \mid \mathbf{x}_\mathcal{T}, \mathbf{x}_\mathcal{C}, \mathbf{y}_\mathcal{C}\right) \quad (15)
$$
$$
p_1\left(\pi(\mathbf{y}_\mathcal{T}(1)) \mid \pi(\mathbf{x}_\mathcal{T}), \mathbf{x}_\mathcal{C}, \mathbf{y}_\mathcal{C}\right) = p_1\left(\mathbf{y}_\mathcal{T}(1) \mid \mathbf{x}_\mathcal{T}, \mathbf{x}_\mathcal{C}, \mathbf{y}_\mathcal{C}\right) \quad (16)
$$

We can see that the distribution $p_1$ satisfies the properties in Definition 1 and Definition 2 of the main text.

Let $\mathcal{S}_c$ and $\mathcal{S}_t$ denote *context score* and *target score* of $\mathcal{S}_{\boldsymbol{\theta}}(\mathbf{y}_\mathcal{C}(t), \mathbf{y}_\mathcal{T}(t), \mathbf{x}_\mathcal{C}, \mathbf{x}_\mathcal{T}, t)$. From the descirption in the Section 3.1 of the the main text, the conditional score function $\nabla_{\mathbf{y}_\mathcal{T}(t)} \log p_t\left(\mathbf{y}_\mathcal{T}(t) \mid \mathbf{x}_\mathcal{T}, \mathcal{C}\right)$ we are interested in is estimated by

$$
\nabla_{\mathbf{y}_\mathcal{T}(t)} \log p_t\left(\mathbf{y}_\mathcal{T}(t) \mid \mathbf{x}_\mathcal{T}, \mathcal{C}\right) = \mathcal{S}_t - \frac{1}{2r^2} \nabla_{\mathbf{y}_\mathcal{T}(t)} \|\mathbf{y}_\mathcal{C} - (\hat{\mathbf{y}}_\mathcal{C}(t) - \beta_t^2 \mathcal{S}_c)/\alpha_t\|_2^2 \quad (17)
$$

Since $\mathcal{S}_{\boldsymbol{\theta}}$ satisfies permutation equivariant function for data pairs, the two term on the right hand in above equation will remain unchanged for the permutation of context data pairs, and change in accordance with the permutation of target pairs. The reverse-time SDE is,

$$
\mathrm{d}\mathbf{y}_\mathcal{T}(t) = \underbrace{\left[\mu(t)\mathbf{y}_\mathcal{T}(t) - \sigma(t)^2 \nabla_{\mathbf{y}_\mathcal{T}(t)} \log p_t\left(\mathbf{y}_\mathcal{T}(t) \mid \mathbf{x}_\mathcal{T}, \mathcal{C}\right)\right]}_{:=f} \mathrm{d}t + \sigma(t)\mathrm{d}\hat{\mathbf{w}}_t, \quad t \in [0,1], \quad (18)
$$

Apparently, the value of $f$ also follows these properties for permutation of context pairs and target pairs. For simplicity, we use $p_t$ to denote $p_t\left(\mathbf{y}_\mathcal{T}(t) \mid \mathbf{x}_\mathcal{T}, \mathcal{C}\right)$. Now the probability density $p_t$ evolves can be written as following by the famous Fokker-Planck equation (Risken & Risken, 1996),

$$
\frac{\partial p_t}{\partial t} = -\sum_{i=1}^{T} \frac{\partial f_i p_t}{\partial y_i(t)} + \frac{\sigma(t)}{2} \sum_{i=1}^{T} \frac{\partial^2 p_t}{\partial y_i(t)^2} \quad (19)
$$

Since the $p_1$ satisfies the properties in Definition 1 and Definition 2, the permutation in either the context set or the target set order does not affect the value of the right-hand side of Eq. (19) at time 1. Therefore, we have,

$$\frac{\partial p_1\left(\mathbf{y}_{\mathcal{T}}(1) \mid \mathbf{x}_{\mathcal{T}}, \pi(\mathbf{x}_{\mathcal{C}}), \pi(\mathbf{y}_{\mathcal{C}})\right)}{\partial t} = \frac{\partial p_1\left(\mathbf{y}_{\mathcal{T}}(1) \mid \mathbf{x}_{\mathcal{T}}, \mathbf{x}_{\mathcal{C}}, \mathbf{y}_{\mathcal{C}}\right)}{\partial t} \tag{20}$$

$$\frac{\partial p_1\left(\pi(\mathbf{y}_{\mathcal{T}}(1)) \mid \pi(\mathbf{x}_{\mathcal{T}}), \mathbf{x}_{\mathcal{C}}, \mathbf{y}_{\mathcal{C}}\right)}{\partial t} = \frac{\partial p_1\left(\mathbf{y}_{\mathcal{T}}(1) \mid \mathbf{x}_{\mathcal{T}}, \mathbf{x}_{\mathcal{C}}, \mathbf{y}_{\mathcal{C}}\right)}{\partial t} \tag{21}$$

Then at sufficiently small time changes $t' = 1 - \Delta t$,

$$p_{t'}\left(\mathbf{y}_{\mathcal{T}}(t') \mid \mathbf{x}_{\mathcal{T}}, \pi(\mathbf{x}_{\mathcal{C}}), \pi(\mathbf{y}_{\mathcal{C}})\right) \tag{22}$$

$$= p_1\left(\mathbf{y}_{\mathcal{T}}(1) \mid \mathbf{x}_{\mathcal{T}}, \pi(\mathbf{x}_{\mathcal{C}}), \pi(\mathbf{y}_{\mathcal{C}})\right) - \frac{\partial p_1\left(\mathbf{y}_{\mathcal{T}}(1) \mid \mathbf{x}_{\mathcal{T}}, \pi(\mathbf{x}_{\mathcal{C}}), \pi(\mathbf{y}_{\mathcal{C}})\right)}{\partial t}\Delta t$$

$$= p_1\left(\mathbf{y}_{\mathcal{T}}(1) \mid \mathbf{x}_{\mathcal{T}}, \mathbf{x}_{\mathcal{C}}, \mathbf{y}_{\mathcal{C}}\right) - \frac{\partial p_1\left(\mathbf{y}_{\mathcal{T}}(1) \mid \mathbf{x}_{\mathcal{T}}, \mathbf{x}_{\mathcal{C}}, \mathbf{y}_{\mathcal{C}}\right)}{\partial t}\Delta t$$

$$= p_{t'}\left(\mathbf{y}_{\mathcal{T}}(t') \mid \mathbf{x}_{\mathcal{T}}, \mathbf{x}_{\mathcal{C}}, \mathbf{y}_{\mathcal{C}}\right) \tag{23}$$

We can see that the $p_{t'}$ also satisfies the properties in Definition 1 and Definition 2. By mathematical induction, the following equation holds for any $t \in [0, 1]$,

$$p_t\left(\mathbf{y}_{\mathcal{T}}(t) \mid \mathbf{x}_{\mathcal{T}}, \pi(\mathbf{x}_{\mathcal{C}}), \pi(\mathbf{y}_{\mathcal{C}})\right) = p_t\left(\mathbf{y}_{\mathcal{T}}(t) \mid \mathbf{x}_{\mathcal{T}}, \mathbf{x}_{\mathcal{C}}, \mathbf{y}_{\mathcal{C}}\right) \tag{24}$$

By the same token, we can prove,

$$p_t\left(\pi(\mathbf{y}_{\mathcal{T}}(t)) \mid \pi(\mathbf{x}_{\mathcal{T}}), \mathbf{x}_{\mathcal{C}}, \mathbf{y}_{\mathcal{C}}\right) = p_t\left(\mathbf{y}_{\mathcal{T}}(t) \mid \mathbf{x}_{\mathcal{T}}, \mathbf{x}_{\mathcal{C}}, \mathbf{y}_{\mathcal{C}}\right) \tag{25}$$

## B  RELATED WORKS

**Neural Processes.** The CNP (Garnelo et al., 2018a) is a groundbreaking approach that combined neural networks with stochastic processes, enabling learning of a suitable prior from multiple related tasks. However, it has been found that CNPs often produce irrelevant predictions and are underfit to the data distribution. The NP (Garnelo et al., 2018b) overcomes this issue by introducing a global latent variable and using variational inference. The ANP (Kim et al., 2019) further uses attention mechanics to improve model expressiveness. Recent works (Vaughan et al., 2021; Foong et al., 2020; Bruinsma et al., 2021; Markou et al., 2022) have also introduced the inductive bias of translation equivariance to NP models by defining convolution operations on sets. These approaches allow for better extrapolation performance in scenarios with stationary processes. Additionally, Nguyen & Grover (2022) and Bruinsma et al. (2023) have employed autoregressive constructions to improve the model's ability to handle complex distributions. In recent studies, Rastogi et al. (2022) and Feng et al. (2022) have introduced induced point-based approaches to NPs, with the goal of achieving computationally efficient inference. In contrast, our contribution lies in the integration of trainable induced points into the score network. This innovation not only reduces the inference workload but also enables the training of score matching without explicit partitioning of the context and target sets. Additionally, the Neural Diffusion Process (NDP) introduced by (Dutordoir et al., 2023) is relevant to our research, given its emphasis on sampling from the function's distribution. However, our primary innovation centers around the development of an efficient method for estimating conditional scores for target sets, circumventing the complexity associated with likelihood calculations. This represents a significant advancement in our work.

**Score-based generative models.** Score matching (Song et al., 2020a; Pang et al., 2020; Vincent, 2011) is a method that was originally used to train energy-based models (LeCun et al., 2007; Song & Kingma, 2021), which allowed to avoid intractable partition functions. Song & Ermon (2019) introduced a new type of generative model called the score-based generative model (SGM) that uses a neural network to parametrize the score function of a perturbed sample in order to model the distribution. Subsequent work (Song & Ermon, 2020; Song et al., 2020b; Ho et al., 2020) further developed the theory of SGMs and they have proven to be highly successful for generating data in a variety of domains (Song et al., 2021; Vahdat et al., 2021; Saharia et al., 2022). One of the advantages of SGMs is that they do not rely on adversarial training (Goodfellow et al., 2020) or strict constraints on model architectures (Kingma & Dhariwal, 2018), making the training and sampling process more stable and flexible compared to other generative models. Additionally, Pavasovic et al. (2022) utilized score matching techniques for learning prior for function space Bayesian neural networks to achieve meta-learning.

## C    EXPERIMENTAL AND IMPLEMENTATION DETAILS

In the following we list the data set and implementation details of the experiment. All experiments are implemented by python and PyTorch 1.10.0, running on a single RTX 3090 GPU.

### C.1    SYNTHETIC EXPERIMENTS

**Datasets.** We build the meta-datasets for regression from the following three settings:

*1) GPs with a single kernel.* we generate functions from GPs with RBF, Periodic and Matern kernels, respectively, to build three meta-datasets. The length scale of RBF kernels is set to 0.2. The length scale and periodicity of Periodic kernels are set to 1 and 0.5. The length scale and smoothness parameter of Matern kernels are set to 0.2 and 1.5. These datasets are used to verify the model's ability to fit a ground truth GP.

*2) GPs with varying kernels.* we generate functions from GPs with three above kernels (i.e., RBF, Periodic, and Matern kernels) and combine them into one meta-dataset. This experiment requires the model to learn to distinguish samples from different kernels.

*3) GPs with varying kernel hyperparameters.* we generate functions from GPs with Matern kernels, with a length scale of $l \sim U[0.01, 0.3]$. This experiment evaluates whether the model could represent a group of GPs with varying kernel hyperparameters.

In each experiment setting, we first draw different functions from the Gaussian Process prior with specific kernels, then choose 128 random locations to evaluate, and sample 10% evaluation as context sets. The remainder is the target points for regression. The testing sets contain 10k functions for each kernel.

**Hyperparameters.**

Number of training epochs: 1000

Batch size: 128

Learning rate: 2e-4

Number of inducing points: 16

Hidden dimension: 64

Number of attention head: 2

Number of ISAB block: 6

Value of $r$ for $q(\mathbf{y}_\mathcal{C} \mid \hat{\mathbf{y}}_\mathcal{C}(t), \mathbf{y}_\mathcal{T}(t), \mathbf{x}_\mathcal{C}, \mathbf{x}_\mathcal{T})$: 1

### C.2    EEG REGRESSION

**Datasets.** We utilize the publicly accessible EEG dataset from the UCI Datasets website for the experiments. The 121 subjects in the filtered dataset each have multiple trial results. We set aside 20 subjects as the testing set and the remaining subjects' data for training. Each time series consists of 256 measurements evenly spread over 64 EEG channels, of which we retain the seven channels containing the electrodes FZ, F1, F2, F3, F4, F5 and F6, following Markou et al. (2022).

In order to verify whether the model can capture the correlation between channels and recover the missing voltage data, we consider it as a two-dimensional regression problem, whose input is the corresponding time and channel index and output is the corresponding voltage value, we set four of the channels (F3, F4, F5 and F6) as observable context points, for the other three channels we comply with the three experimental settings in the main text, divide some points to merge with the above four channels as the final context set, and the remaining points in these three channels as the target set.

*1) Interpolation:* we randomly select a number of the 256 points uniformly to be target points and use the remaining points as context points.

*2) Reconstruction:* we chose a window of a specific size in the time range and use the points outside the window as context to regress the points inside the window.

*3) Forecasting:* we randomly select a time point within the time range, with the one before this point as the context point, and the one after it as the forecasting target.

For *Reconstruction* and *Forecasting*, we keep 50% of the measurements on these three channels as context, while for the *Interpolation* experiments we keep only 10%, We also performed standardized preprocessing on the data.

**Hyperparameters.**

Number of training epochs: 500

Batch size: 128

Learning rate: 2e-4

Number of inducing points: 64

Hidden dimension: 256

Number of attention head: 4

Number of ISAB block: 6

Value of $r$ for $q(\mathbf{y}_\mathcal{C} \mid \hat{\mathbf{y}}_\mathcal{C}(t), \mathbf{y}_\mathcal{T}(t), \mathbf{x}_\mathcal{C}, \mathbf{x}_\mathcal{T})$: 0.1

## C.3 IMAGES ON GRIDS

**Datasets.** Image data can be interpreted as being generated from a stochastic process, where each image can be thought of as a unique function, and predicting the pixel values can be cast as a 2D regression task mapping a 2D pixel location to its pixel intensity. Specifically, we adopt MNIST and CelebA with $32 \times 32$ size the follow the standard train/test split. The coordinates and pixel intensities were both rescaled to the range $[-1, 1]$.

In the testing phase, we tried two meta-regression settings. One is to randomly select 10% of the pixels as the background for recovering the remaining target pixels, and the other is to mask half of the image and use the NP model to predict the pixels in the other half of the coordinates. Obviously, this case is more difficult to predict and the uncertainty of pixel values is higher.

**Hyperparameters.**

*MNIST*

Number of training epochs: 800

Batch size: 128

Learning rate: 2e-4

Number of inducing points: 64

Hidden dimension: 256

Number of attention head: 4

Number of ISAB block: 6

Value of $r$ for $q(\mathbf{y}_\mathcal{C} \mid \hat{\mathbf{y}}_\mathcal{C}(t), \mathbf{y}_\mathcal{T}(t), \mathbf{x}_\mathcal{C}, \mathbf{x}_\mathcal{T})$: 1

*CelebA32*

Number of training epochs: 500

Batch size: 128

Learning rate: 2e-4

Number of inducing points: 128

Hidden dimension: 256

Number of attention head: 4

Number of ISAB block: 6

Value of $r$ for $q(\mathbf{y}_{\mathcal{C}} \mid \hat{\mathbf{y}}_{\mathcal{C}}(t), \mathbf{y}_{\mathcal{T}}(t), \mathbf{x}_{\mathcal{C}}, \mathbf{x}_{\mathcal{T}})$: 1

### C.4 FLOW FIELDS ON MESHES

**Datasets.** The CYLINDERFLOW dataset simulates the flow of water around a cylinder on a fixed 2D Eulerian mesh and contains 1200 flow field simulation trajectories, each of which consists of 600 continuous time observations with a total number of evaluation nodes between 1735 and 2036 with a mean of approximately 1888. Following the original partition, we use 1000 trajectories for training and the remaining for testing.

Our target is to learn the data distribution of the velocity fields of every node given its position, time and node type. To introduce time information, we encode it as 600 time-steps between -1 and 1 uniformly. Then, the time encoding and static node type information (such as fluid nodes, wall nodes and inflow/outflow boundary nodes) are both concatenating to the coordinates of every point, resulting in 4 channels input. In addition, to match our inferring strategy, we put every adjacent 6 time-steps together for all evaluation nodes to train SNPs. We evaluate these models on completing the missing nodes spatially with a 98% mask rate. We also demonstrate that our SNPs can also work well with additional masks temporally, which can be used to solve PDEs.

**Hyperparameters.**

Number of training epochs: 20

Batch size: 128

Learning rate: 2e-4

Number of inducing points: 64

Hidden dimension: 256

Number of attention head: 4

Number of ISAB block: 6

Value of $r$ for $q(\mathbf{y}_{\mathcal{C}} \mid \hat{\mathbf{y}}_{\mathcal{C}}(t), \mathbf{y}_{\mathcal{T}}(t), \mathbf{x}_{\mathcal{C}}, \mathbf{x}_{\mathcal{T}})$: 1

### C.5 CLIMATE DATA ON MANIFOLDS

**Datasets.** We adopt temperature measurements from the ERA5 dataset over the last 40 years. We follow Dupont et al. (2021) preprocess the data. The dataset under consideration pertains to monthly averaged surface temperature measurements obtained from hourly reanalysis records, obtained via a grid of 721 latitudes and 1440 longitudes spanning the globe. Each temperature measurement is recorded at a height of 2 meters above the surface of the Earth, resulting in a data point that contains a temperature reading at each of the 721 x 1440 grid points. To reduce the dataset's size, we employ a subsampling strategy that reduces the grid by a factor of 16, resulting in a smaller grid of size 46 x 90. We extract 24 grids from this reduced grid for each hour of the day, resulting in a total of 12096 datapoints for the entire period of January 1979 to December 2020.

Handling missing climate data is a critical step in ensuring the accuracy of weather forecasts and climate projections. Thereby we conducte two experiments to evaluate the effectiveness of different NP models: The first involves randomly selecting a small number of temperature measurements as context, which are then used to predict most of the temperature values at the remaining positions. Since the distribution of weather stations may be uneven in many regions, with some areas having a dense network of stations while others have very few or none at all. The second setting involves regressing a large region of missing temperature measurements, requiring the model to make predictions from more distant contextual points. This scenario is relevant to situations where entire regions may be missing data due to technical limitations or natural phenomena. We divide this dataset and build a train set with 8510 measurements and a test set with 2420 measurements. We also normalize the data to lie in $[0, 1]$.

**Hyperparameters.**

Number of training epochs: 300

Batch size: 128

Learning rate: 2e-4

Number of inducing points: 64

Hidden dimension: 256

Number of attention head: 4

Number of ISAB block: 6

Value of $r$ for $q(\mathbf{y}_\mathcal{C} \mid \hat{\mathbf{y}}_\mathcal{C}(t), \mathbf{y}_\mathcal{T}(t), \mathbf{x}_\mathcal{C}, \mathbf{x}_\mathcal{T})$: 1

## C.6 BASELINE SETTINGS

To implement NPs, ANPs, and ConvNPs and reproduce the results, we employ the *Neural Process Family* repository (Dubois et al., 2020), and set the hyperparameters to the recommended default values. We have retrained the model on all experimental datasets for metric calculation and visualization. For NPs and ANPs, we use ELBO as the objective function for training. We find that ELBO does not work well for ConvNPs, so we follow Foong et al. (2020) to use Monte-Carlo approximation to approximate the likelihood of computing the objective as the training criterion.

## C.7 METRIC CALCULATING

We employ two main metrics to evaluate the performance of our models: mean squared error (MSE) and negative log-likelihood (NLL). The MSE is calculated by computing the squared difference between the predicted and true values at each target point and taking the average over all points. This metric provides a measure of the average discrepancy between the predicted and true values. For NPs and ANPs, we follow Foong et al. (2020) and use importance weighting to estimate their NLL. For ConvNPs that are not trained with ELBO, the importance weights may be ill-suited, thereby we use the Monte Carlo approximation to estimate their NLL. Similar to the score-based generative model, SNPs can compute the exact likelihood by constructing an equation for the instantaneous change of variables and solving an ordinary differential equation (ODE). Specifically, for the following reverse-time SDE,

$$d\mathbf{y}_\mathcal{T}(t) = \left[\mu(t)\mathbf{y}_\mathcal{T}(t) - \sigma(t)^2 \nabla_{\mathbf{y}_\mathcal{T}(t)} \log p_t\left(\mathbf{y}_\mathcal{T}(t) \mid \mathbf{x}_\mathcal{T}, \mathcal{C}\right)\right] dt + \sigma(t) d\hat{\mathbf{w}}_t, \quad t \in [0, 1], \quad (26)$$

there exists an ODE with same marginal probability densities $\{p_t\left(\mathbf{y}_\mathcal{T}(t) \mid \mathbf{x}_\mathcal{T}, \mathcal{C}\right)\}_{t \in [0,1]}$ corresponding to it,

$$d\mathbf{y}_\mathcal{T}(t) = \underbrace{\left[\mu(t)\mathbf{y}_\mathcal{T}(t) - \frac{1}{2}\sigma(t)^2 \nabla_{\mathbf{y}_\mathcal{T}(t)} \log p_t\left(\mathbf{y}_\mathcal{T}(t) \mid \mathbf{x}_\mathcal{T}, \mathcal{C}\right)\right]}_{:=\mathcal{G}} dt, \quad t \in [0, 1], \quad (27)$$

The theory in the neural ODE Chen et al. (2018) derive another ODE describing the change in log probability. It can be written as follows,

$$\frac{\partial \log p_t\left(\mathbf{y}_\mathcal{T}(t) \mid \mathbf{x}_\mathcal{T}, \mathcal{C}\right)}{\partial t} = -\operatorname{tr}\left(\frac{d\mathcal{G}}{d\mathbf{y}_\mathcal{T}(t)}\right) \quad (28)$$

Since Eq. (27) establishes a one-to-one correspondence between $\mathbf{y}_\mathcal{T}$ and $\mathbf{y}_\mathcal{T}(1) \sim \mathcal{N}(\mathbf{0}, \mathbf{I})$, and the likelihood of $\mathbf{y}_\mathcal{T}(1)$ is easy to obtain, we can calculate the likelihood of target outputs by solving Eq. (28). In order to reduce the computational consumption of the traces, we further adopt Hutchinson's trace estimator (Grathwohl et al., 2018). This technique involves projecting the matrix onto a random vector, significantly reducing computation using automatic differentiation libraries. While this method provides an unbiased estimate of the trace, there exists some inaccuracies in the likelihood computation. We mitigate this by performing multiple runs to approximate the true likelihood accurately. Specifically, we conduct five averages to present results in the main text. For more details of the algorithm, please refer to Song et al. (2020b); Grathwohl et al. (2018).

## C.8 INDUCED SET ATTENTION BLOCKS

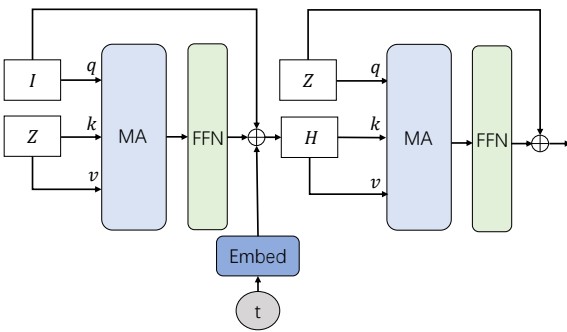

Figure 6: Diagram of ISAB architecture. *MA* is the multihead attention layer and *FFN* is the forward-feed network. *Embed* is the embedding layer for time information. *I* is the trainable induced points

## D RESULTS OF ABLATION EXPERIMENTS

### D.1 ABLATION FOR ISAB

Table 6: Comparison of ISAB and Vanilla Attention on EEG

|  | Interpolate | | Reconstruction | | Forecasting | | |
| Model | MSE | NLL | MSE | NLL | MSE | NLL | MACs (G) |
| --- | --- | --- | --- | --- | --- | --- | --- |
| ISAB | **0.12** | -2.37 | **0.11** | -2.39 | **0.22** | -2.35 | 0.58 |
| Vanilla Attention | 0.13 | **-2.38** | 0.15 | **-2.41** | 0.24 | **-2.38** | 0.58 |

Table 7: Comparison of ISAB and Vanilla Attention on ERA5

|  | Large-region missing | | Random missing | | |
| Model | MSE($10^{-2}$) | NLL | MSE($10^{-2}$) | NLL | MACs (G) |
| --- | --- | --- | --- | --- | --- |
| ISAB | **0.05** | **-5.19** | **0.06** | **-5.02** | **7.77** |
| Vanilla Attention | 0.10 | -4.49 | 0.09 | -4.83 | 60.30 |

### D.2 ABLATION FOR RUNTIME TRADEOFF

Table 8: Runtime/performance under varying NFE on EEG

| Method | SNP(NFE=1000) | SNP(NFE=500) | SNP(NFE=100) | SNP(NFE=50) | NP | ANP | ConvNP |
| --- | --- | --- | --- | --- | --- | --- | --- |
| Interpolate (MSE) | 0.119 | 0.139 | 0.161 | 0.185 | 0.351 | 0.206 | 0.295 |
| Reconstruction (MSE) | 0.115 | 0.140 | 0.169 | 0.207 | 0.320 | 0.350 | 0.297 |
| Forecast (MSE) | 0.224 | 0.261 | 0.335 | 0.364 | 0.417 | 0.474 | 0.419 |
| Time (s/sample) | 0.451 | 0.225 | 0.045 | 0.023 | 0.001 | 0.006 | 0.009 |

# E    ADDITIONAL QUALITATIVE RESULTS

In this section, we present some additional qualitative visualization results in experiments.

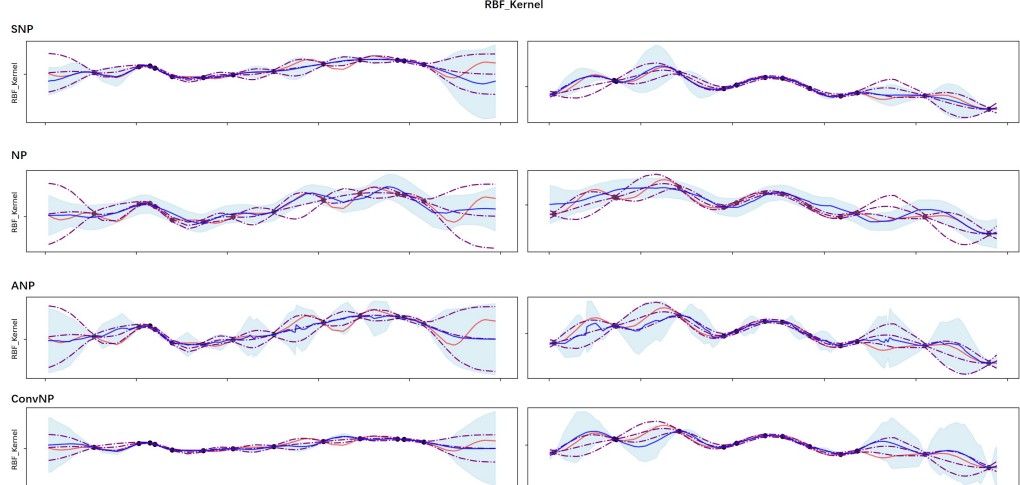

Figure 7: Qualitative evaluation on functions from GPs with RBF kernels. The blue lines and the shaded blue area denote the predictive $\mu \pm 2\sigma$. The red lines is the function sample. Purple dash–dot lines are the ground-truth GP mean and $\mu \pm 2\sigma$.

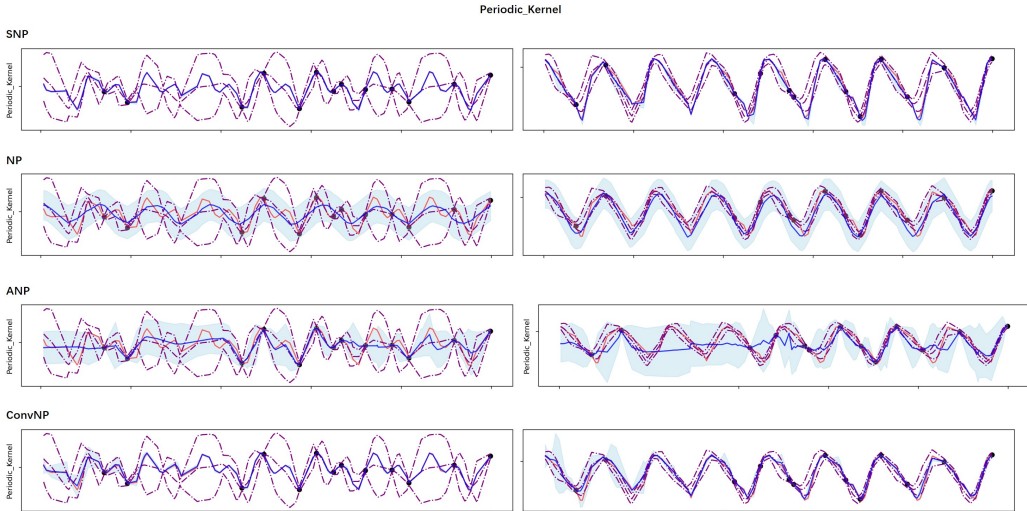

Figure 8: Qualitative evaluation on functions from GPs with Periodic kernels. The blue lines and the shaded blue area denote the predictive $\mu \pm 2\sigma$. The red lines is the function sample. Purple dash–dot lines are the ground-truth GP mean and $\mu \pm 2\sigma$.

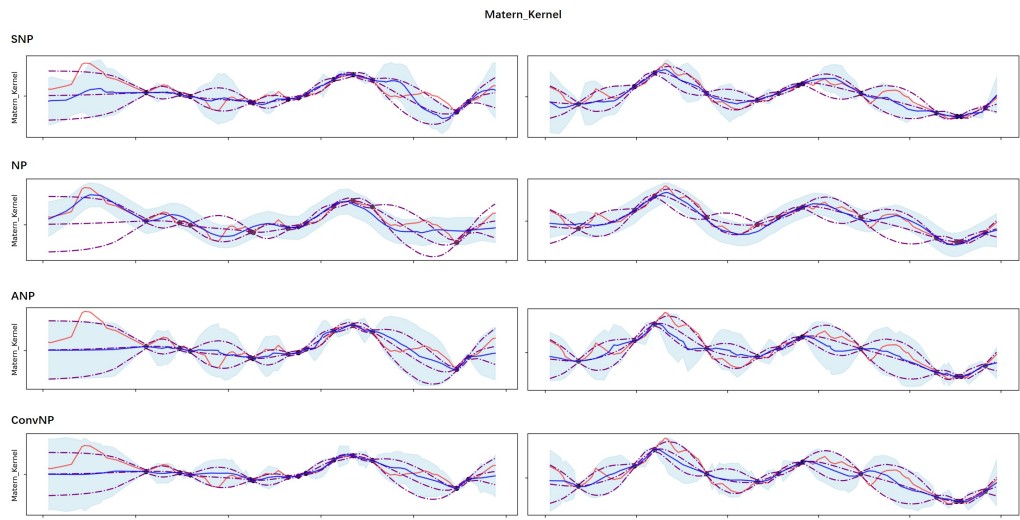

Figure 9: Qualitative evaluation on functions from GPs with Matern kernels. The blue lines and the shaded blue area denote the predictive $\mu \pm 2\sigma$. The red lines is the function sample. Purple dash–dot lines are the ground-truth GP mean and $\mu \pm 2\sigma$.

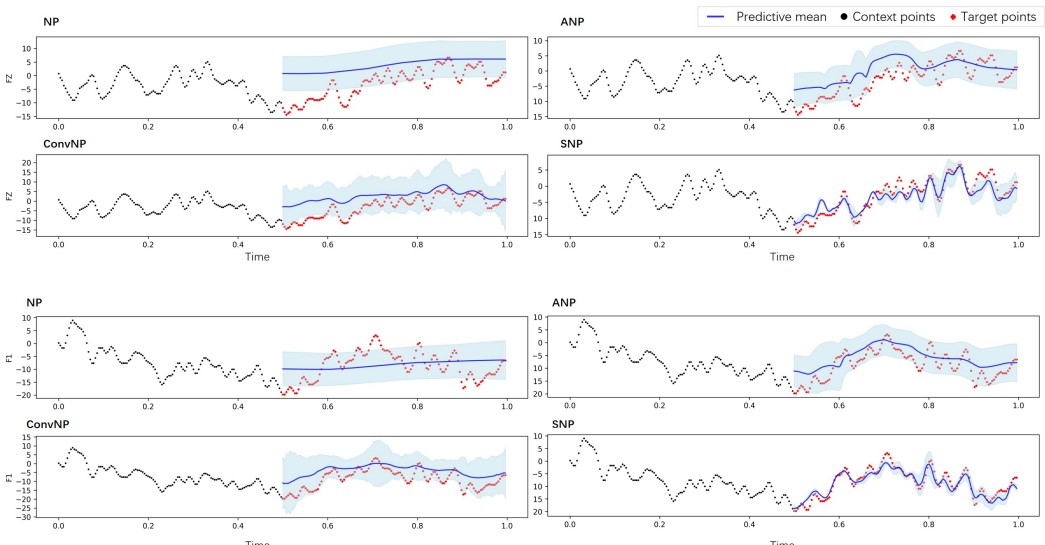

Figure 10: Qualitative evaluation of *Forecasting* on EEG. The blue lines and the shaded blue area denote the predictive mean and standard deviation.

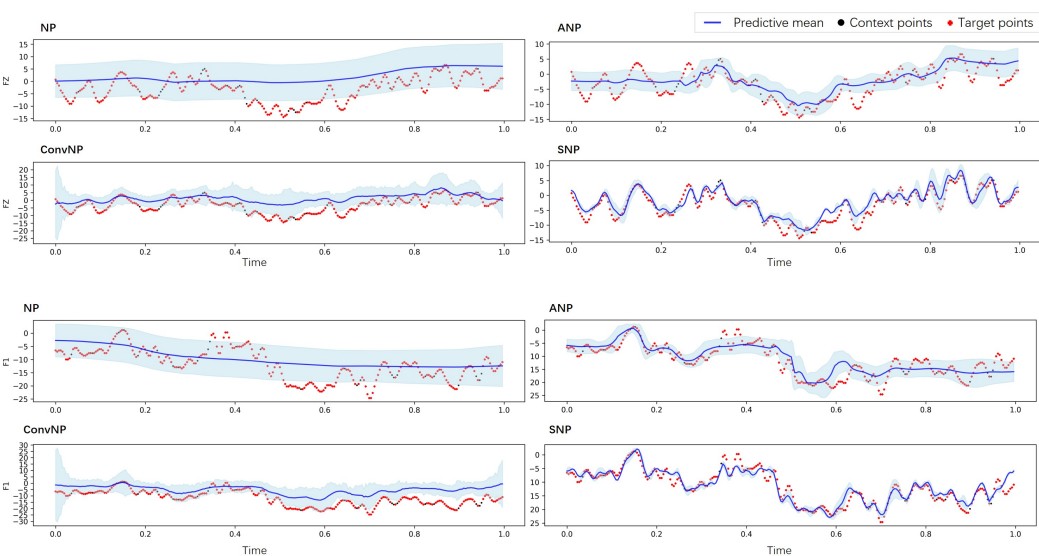

Figure 11: Qualitative evaluation of *Interpolation* on EEG. The blue lines and the shaded blue area denote the predictive mean and standard deviation.

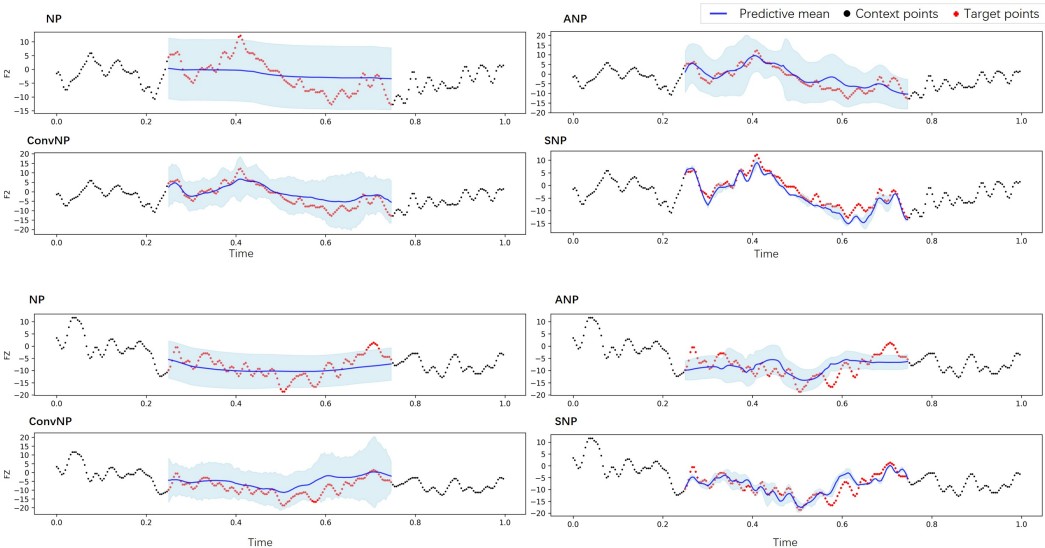

Figure 12: Qualitative evaluation of *Reconstruction* on EEG. The blue lines and the shaded blue area denote the predictive mean and standard deviation.

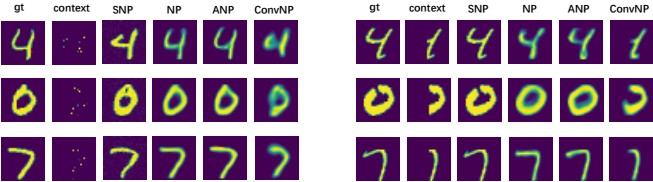

Figure 13: Qualitative evaluation of MNIST.

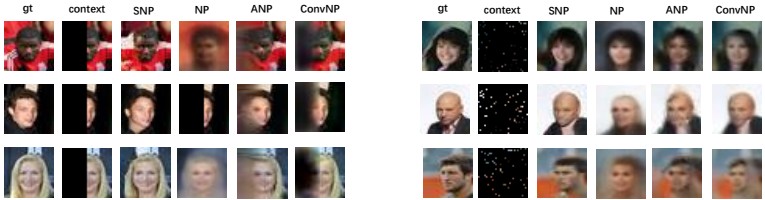

Figure 14: Qualitative evaluation of CelebA32.

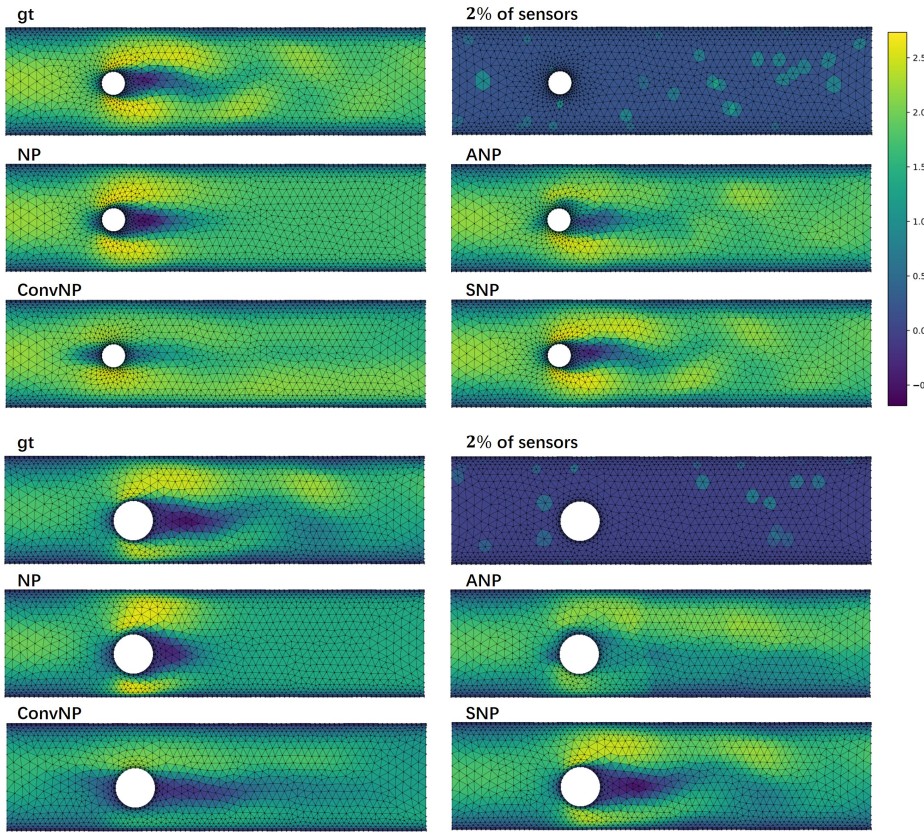

Figure 15: Qualitative evaluation of under 2% of sensors on CYLINDERFLOW. The color map shows the x-component of the velocity field.

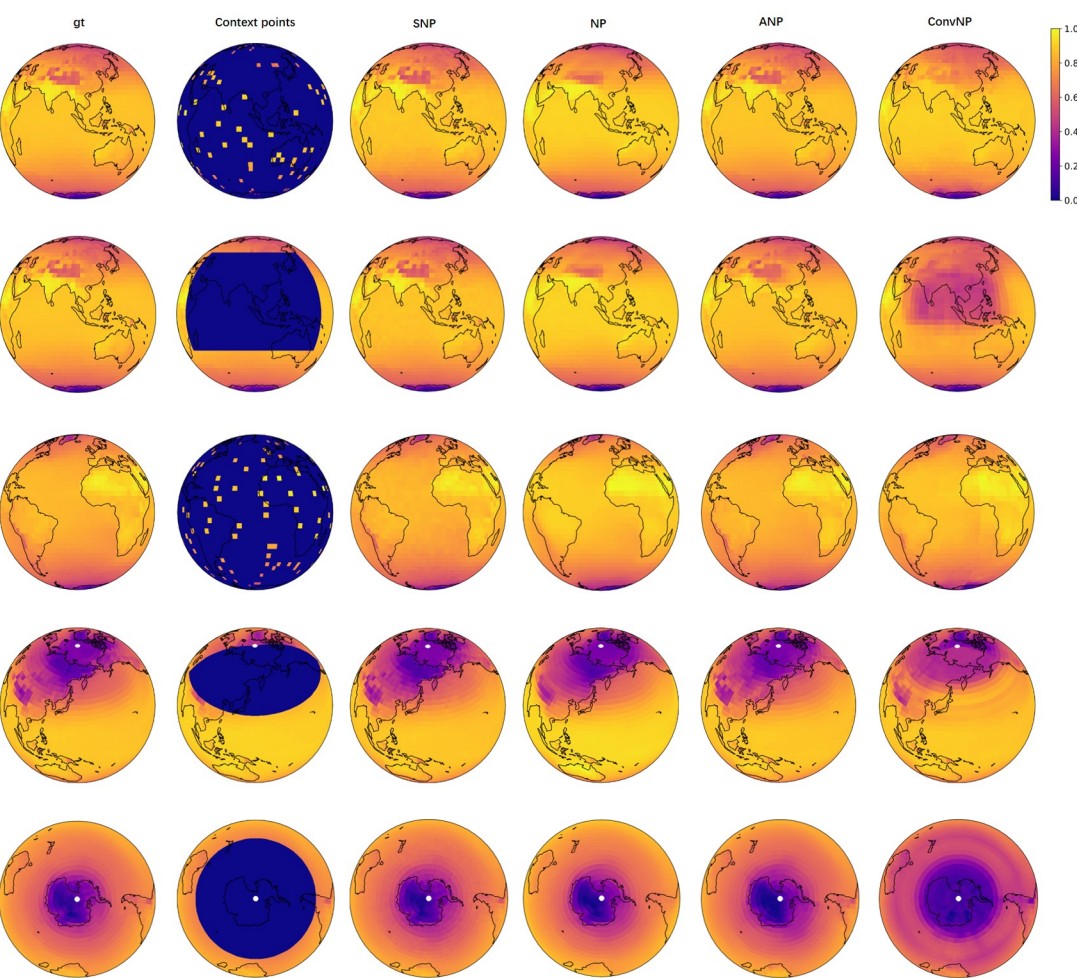

Figure 16: Qualitative evaluation of ERA5 under varying types of measurement missing and varying perspectives.

