# OpenReview forum: "Score-based Neural Processes"
_ICLR.cc/2024/Conference — ICLR 2024 Conference Withdrawn Submission_

### Official Review · Reviewer_Gnhg · 2023-10-24

**Soundness:** 2 fair
**Presentation:** 3 good
**Contribution:** 2 fair
**Rating:** 5
**Confidence:** 4

**Summary:**

The main objective of this work is to address the issue of intractable log-likelihood in predictive functions within the neural process model. The key contribution of this study is the utilization of the score-matching technique in the output space. To evaluate the effectiveness of this approach, experiments were conducted using both synthetic and real-world datasets.

**Strengths:**

This motivation of this work is clear and the use of score-matching can be beneficial to the NP family.
The manuscript is easy to follow, and the experiments are complete in evaluation.

**Weaknesses:**

**1. About the novelty.**

It seems there already exist a collection of methods combining the score-matching in neural processes or other meta learning models [1-3]. Advantages of this work over previous ones are not clear in some sense.

[1] Dutordoir V, Saul A, Ghahramani Z, et al. Neural diffusion processes[J]. arXiv preprint arXiv:2206.03992, 2022.

[2] Pavasovic, K. L., Rothfuss, J., & Krause, A. (2022). MARS: Meta-learning as score matching in the function space. arXiv preprint arXiv:2210.13319.

[3] Mathieu E, Dutordoir V, Hutchinson M J, et al. Geometric Neural Diffusion Processes[J]. arXiv preprint arXiv:2307.05431, 2023.

**2. About the experimental evaluation.**

The use of ISAB modules inevitably increase model complexity in comparison to vanilla NPs, where MLPs are used. I notice that authors claim the comparable model complexity are set in running different baselines. However, the encoder design or model complexity is not well explained.

Another concern lies in the set-up about the context and the target. On page 4, it seems the context and the target are disjoint while the context is the subset of the target in general NPs.

Meanwhile, the evaluation in image completion is not clear about the number of context points.

**Questions:**

See the weakness part.

---

### Official Review · Reviewer_RB5P · 2023-10-31

**Soundness:** 2 fair
**Presentation:** 2 fair
**Contribution:** 2 fair
**Rating:** 3
**Confidence:** 3

**Summary:**

In this paper, the authors propose score-based neural processes, a type of meta-learning model which combines neural processes with score-based modelling.
They propose corrupting the output values with a Gaussian corruption process and using an attention-based neural network to denoise them.
They suggest an approximation for obtaining a conditional score from a learnt joint score (theorem 1), and use this for conditional sampling.
In experiments across a range of different regression problems, the authors find that their score-based NP compares favourably to existing approaches.

**Strengths:**

In my view, the strengths of this work are as follows:

__Interesting idea for a flexible model:__
Combining score-based models with NPs is an interesting approach for building a flexible meta-learning model.
This appears to be a positive aspect of this work, although I have substantial doubts about its originality and significance, given that Dutordoir et al. (2023) already introduced a method which appears to be extremely similar.
Therefore, while the idea presented in this paper is very interesting, I also have some doubts about the originality of this submission (see weaknesses section below).

__Range of experiments:__
The authors present a range of experiments including synthetic data, EEG data, images on grids, fluid flows and weather data, to assess the performance of their method.
The score-based NP proposed by the authors seems to perform well across these data modalities (although the quality of the baselines used in some of these experiments does not seem to be particularly good; see weaknesses section below).
The variety in these experiments is a strong point of this work.

**Weaknesses:**

In my assessment, the main weaknesses of this paper are as follows:

__Similarity to Dutordoir et al. (2023):__
The present work appears highly similar to the work of Dutordoir et al. (2023), who introduced Neural Diffusion Processes (NDPs).
In an NDP, the target and context output variables (and optionally also the input variables) are corrupted according to a diffusion process, and then denoised using a reverse process where the score function is parameterised using an attention based network whose parameters are learnt from the data.
The present work does not seem any different from Dutordoir et al. in this respect (see for example eq. 10, which shows that the same denoising score matching loss is used as the training objective).
The main difference of this work to that of Dutordoir et al. seems to be that in the present work, the authors adopt an unconventional approach for sampling from the diffusion model (see Theorem 1 and eq. 7).
This seems like a minor difference between these works: conditional samples can be drawn from an NDP and associated predictive log-likelihoods can be modelled using standard sampling techniques, without approximations (see questions section below).
Therefore, I do not clearly see how this work differentiates itself from Dutordoir et al. in a substantial way.
Can the authors comment on any aspects that differentiate the present work?

__Theoretical content does not seem fully rigorous:__
The theoretical content of the paper does not seem fully rigorous to me.
For example, in section 3.1 the authors state an "approximate expression" (eq. 7) for the conditional score.
However, the sense in which this expression is approximate is not made rigorous neither in the theorem statement nor in its proof in the apprendix.
I would recommend that the authors clarify the sense of this approximation sign in both the theorem and the proof.
Looking at the proof in appendix A.1, it is clear for example that the authors are approximating an integral with a single monte carlo sample (eq. 13), and then taking the logarithm of this estimate later on (eq. 14).
However, this procedure gives a clearly biased estimate of the score in eq. 14 (first line), which is glossed over using an "approximately equal" sign.
I recommend that the authors should try to make their theoretical claims more precise as, in its current form, the rigour of the paper is not fully satisfactory.

__Quality of baselines:__
It appears that the quality of the baselines to which the SNP is compared to does not appear to be particularly strong, and the extent to which these were carfeully trained and tuned seems insufficient.
For example, the ConvNP appears to perform unexpectedly poorly across a range of the tasks explored.
Further, in some occasions, the SNP performs much better than all other models by a suspiciously large margin.
See for example Table 3, where the SNP appears to outperform all other baselines by around 6-7 nats, which is an extremely large margin.
This raises questions about the quality of the experiments and whether enough care was taken to implement, train and tune the baselines to be competitive with the SNP.
Since there is also no code submitted along with the paper which would be sufficient to check and reproduce these results, the baseline comparisons are not very convincing.


__Summary:__
Due to the above considerations on the originality of the work and the quality of the baselines used in the experiments I have recommended the paper for a score of 3.
However, I invite the authors to comment on this response, particularly with any clarifications they might have on distinguishing features of their work which I may have missed (and remain open to adjusting my score accordingly).

**Questions:**

## Questions

__Why not just learn the conditional score?__
The authors seem to have gone through a convoluted procedure in order to model the conditional score of the corrupted target variables given the context data and the target inputs (see eq. 7).
Instead of making an approximation (which also does not appear to be fully rigorous to me), why not just learn the conditional score of the corrupted target outputs given uncorrupted target outputs?
Concretely, at training time, one could: (1) separate a task into a context and a target set; (2) corrupt the target; (3) compute a score matching denoising loss for the corrupted target outputs given the context outputs (which are provided to the model).
Works such as [Batzolis et al. (2021)](https://arxiv.org/abs/2111.13606) have shown that one can learn the conditional score by passing the conditioning variables as inputs to the score model.

__Why not just sample using the joint score?__
An alternative to the above for conditional sampling is to use the joint score, and a standard method such as in-painting together with a predictor-corrector sampler (Song et al. 2020).
This would allow to sample from the conditionals given just a trained joint score model, without requiring any approximations.

__Unexpectedly poor ConvNP performance:__
In numerous parts of the experimental section, the performance of the ConvNP is significantly worse than that of other models, such as the ANP.
This is surprising given that previous works such as Foong et al. (2020), or Bruinsma et al. (2023) have found that the ConvNP significantly outperforms the ANP (see for example the evaluation tables in the appendix of Bruinsma et al., which are fairly extensive).
Can the authors comment on these results and are they confident about their soundness?


## Recommendations and changes

__Baseline models and implementations:__
Related to the above issue on the unexpectedly poor performance of the ConvNP, in appendix C.6, the authors report that in their experiments they have used the models provided in the Neural Process Family of Dubois et al. (2020).
However, a more extensive set of baseline implementations and carefully picked model parameters (which is also more recent than the repository above) is the [neuralprocesses](https://github.com/wesselb/neuralprocesses) repository.
Two options for cross-checking the validity of their experiments and soundness of their baselines are to use the official code releases of the baselines in question, or the aforementioned repository, which contains more carefully implemented versions of the baselines.

__Using improved samplers:__
In this work, the authors are using an Euler-Maruyama sampler for integrating the reverse SDE.
However, better sample quality has been reported in previous works when using (slightly) more sophisticated sampling schemes, such as predictor-corrector sampling.
Have the authors considered using such samplers?
These seem to be a relatively easy way to obtain higher quality samples with relatively small implementation overheads.

Below are some further minor recommendations to improve the manuscript.
These include more minor clarifications, typographical corrections and a few phrasing suggestions.

- In section 3, the authors write "Regarding the second term, which necessitates estimating the distribution for $\mathbf{y}_{\mathcal{T}}$..."
It is not clear what the authors mean to say with this expression and the following equation.
Would it be possible to clarify on this point?
- In section 3.1 the authors write "Remarkably, this variance is intricately linked to the time progression of the perturbation process within the VP SDE."
It is not clear to me what the authors mean to say here.
Would it be possible to clarify this point here and in the main text?
More generally, in this paragraph, the authors are trying to justify their approximation in eq. 7, but the argument provided is not convincing.
- Typo change "Previous study (...)" to "Previous studies (...)."
Is the $q$ distribution defined in eq. 8 an approximation to some other quantity, and how is it derived?
- Change "Definition 1 of context permutation invariance..." to "Definition 1..." and "Definition 2 of target permutation equivariance..." to "Definition 2..."
- In the same definitions, correct "then the model is permutation invariant" to "then the model is permutation invariant" and similarly for the equivariant definition.
- Link theorem 2 to the associated proof in the appendix.
- Modify "This represents a significant advancement in our work." to "This is a significant contribution of our work."

---

### Official Review · Reviewer_EWhp · 2023-11-01

**Soundness:** 2 fair
**Presentation:** 3 good
**Contribution:** 2 fair
**Rating:** 5
**Confidence:** 3

**Summary:**

This work proposes to utilise score-based diffusion models for meta-learning. In this regard, the authors adhere to the conventional meta-learning framework, where the training datasets are divided into context and target subsets consisting of input-output data samples. However, instead of following the approach of modelling the conditional distribution of the output in the target subset given the target input and context input-output data samples, a standard approach in Neural Processes, the authors employ a diffusion-based generative modelling approach (Song et al., 2020b) to approximate the distribution of output variables. During the data generation (the reversal of the stochastic diffusion) process, the method relies on the score of the marginal distribution of the output variables to generate the output from noise. As the primary theoretical contribution of the work, the authors derive an approximate representation of the score function for the conditional distribution of (perturbed) output variables. To train a predictor of the score function values, the authors also introduce a computationally efficient neural attention module within the feed-forward architecture of the score-predicting neural network. In the numerical experiments, the authors compare their method to several approaches based on neural processes. They benchmark on both synthetic and real datasets to evaluate the performance and effectiveness of the proposed method.

**Strengths:**

The work addresses a significant area of machine learning.

The approach adopted by the authors appears to be novel within the area of meta learning.

The authors provide a theoretically grounded approximation and factorisation of the score function.

The employed attention mechanism alleviates the quadratic computational complexity w.r.t. the number of data points.

In comparison to various neural processes approaches, the empirical results demonstrate a competitive performance of the proposed method in terms of various metrics.

**Weaknesses:**

I am not sure how the proposed method may fall under the category of neural processes. While neural processes are probabilistic models that explicitly model the conditional distribution of target output variables given target input and context input-output pairs, the proposed method leverages score-based diffusion models for meta-learning, which are fundamentally different from neural processes. Given this, empirical comparisons only involving meta-learning or few-shot approaches based on neural processes are insufficient in my opinion.

The proposed method does have a restriction that it requires the same number of (context+target) data points across datasets, which limits its applicability to tasks involving datasets of equal length, such as multivariate time-series or images. This restriction raises concerns about whether the proposed method (or meta-learning) should be considered as the default approach for tasks considered in the paper such as image inpainting and time-series forecasting. For example, there are competitive diffusion-based methods for image processing. Similarly, in the field of time-series forecasting, there exist few-shot/meta-learning methods that have demonstrated competitiveness and may be better suited for such tasks.

**Questions:**

Can the authors clarify how they compute predictive (log-)likelihood for the proposed SNP approach?

Why is equation 8 chosen to be a Gaussian distribution?

What was the size and dimensionality of synthetic datasets ?